# Efficient Parallelization Layouts
# for Large-Scale Distributed Model Training

**Johannes Hagemann**
Aleph Alpha / Hasso Plattner Institute
`johannes.hagemann@student.hpi.de`

**Samuel Weinbach**
Aleph Alpha
`samuel.weinbach@aleph-alpha.com`

**Konstantin Dobler**
Hasso Plattner Institute
`konstantin.dobler@hpi.de`

**Maximilian Schall**
Hasso Plattner Institute
`maximilian.schall@hpi.de`

**Gerard de Melo**
Hasso Plattner Institute
`gerard.demelo@hpi.de`

## Abstract

Efficiently training large language models requires parallelizing across hundreds of hardware accelerators and invoking various compute and memory optimizations. When combined, many of these strategies have complex interactions regarding the final training efficiency. Prior work tackling this problem did not have access to the latest set of optimizations, such as FLASHATTENTION or sequence parallelism. In this work, we conduct a comprehensive ablation study of possible training configurations for large language models. We distill this large study into several key recommendations for the most efficient training. For instance, we find that using a micro-batch size of 1 usually enables the most efficient training layouts. Larger micro-batch sizes necessitate activation checkpointing or higher degrees of model parallelism and also lead to larger pipeline bubbles. Our most efficient configurations enable us to achieve state-of-the-art training efficiency results over a range of model sizes, most notably a Model FLOPs utilization of 70.5% when training a LLAMA 13B model.

## 1 Introduction

The number of parameters and computational resources spent on training deep neural networks is growing rapidly [1, 3, 14]. The largest models consisting of hundreds of billions of parameters do not even fit onto a single hardware accelerator. Thus, training these models requires various ways of reducing the memory requirements, such as ZeRO [16], activation checkpointing [2], and 3D-parallel (data, tensor, and pipeline parallel) training [13]. 3D parallelism, in particular, has been demonstrated to be effective for the training of Transformer-based large language models (LLMs) with hundreds of billions of parameters [13].

However, training these models efficiently with 3D parallelism requires significant domain expertise and extensive manual effort to determine the ideal configurations. These configurations not only need to combine data, model, and pipeline parallelism most efficiently, but also consider complex interactions with other memory and compute optimizations. FLASHATTENTION [5] in particular has had a notable impact since its release, enabling us to train models at previously impossible degrees of training efficiency. In light of these developments, we conduct a systematic study via a large-scale training efficiency sweep of these interactions. We consider up to 256 GPUs and LLAMA models with up to 65 billion parameters.

Workshop on Advancing Neural Network Training at 37th Conference on Neural Information Processing Systems (WANT@NeurIPS 2023).

We expand on previous work in this direction [13], but include more complex interactions, such as varying the micro-batch size alongside the 3D-parallel configuration. We also investigate the impact of newer methods, such as FLASHATTENTION [5] and sequence parallelism [8], finding that these can affect the optimal training configuration considerably. Our paper provides several actionable insights for efficiently training LLMs. In summary, the contributions of our work are as follows:

- We conduct a large sweep over possible configurations for efficiently training LLMs.
- Our work considers more degrees of freedom in the training configurations than previous work [13] and incorporates important recent techniques such as FLASHATTENTION and sequence parallelism.
- We distill our findings into several, actionable insights that enable a more efficient large-scale training of LLMs.

## 2 Background

Training very large models requires the combination of various techniques for parallelization across devices and other memory and compute optimizations. In the following, we provide an overview of the techniques implemented in our in-house training framework AA-Scaling, which we use to conduct the experiments in this paper. These techniques are also implemented in various other frameworks [28, 10, 19, 17, 25].

**Data Parallelism**   Data parallelism [23] splits the dataset across GPUs during training. Each GPU holds a full model copy, computing loss and gradients for its data shard in parallel. Gradients are then synchronized across devices before weight updates. However, this requires that the model fits entirely within a single GPU's memory. For larger models, we can also shard the optimizer states, gradients, and model parameters across GPUs using techniques like ZeRO or FSDP [16, 27]. However, especially when sharding parameters, this introduces additional communication overhead.

**Tensor Parallelism**   Tensor parallelism splits individual weight matrices across multiple GPUs and computes the matrix multiplication in parallel across them. As each GPU only holds a shard of the full weight matrix, we can fit larger models into memory. For Transformer models, the self-attention and MLP blocks can be parallelized this way with little communication overhead [19]. Due to the natural parallelism of separate attention heads, we only need a single all-reduce operation in both the forward and backward passes. The MLP block similarly requires just a single synchronization in each pass.

**Pipeline Parallelism**   Pipeline parallelism splits the model's layers into subsequent stages across GPUs. Activations are transferred between these stages. As each GPU only holds some of the layers of the model, we can again fit larger models into memory. However, it can introduce "pipeline bubbles" of GPU inactivity due to processing delays. PipeDream [12] is a scheduling algorithm to reduce these by using micro-batches and scheduling their forward and backward computations across pipeline stages. By interleaving forward and backward passes for each micro-batch, PipeDream further reduces memory usage, discarding activations after the specific micro-batch's backward pass.

**3D Parallelism**   As shown by Megatron-LM [19], data, tensor, and pipeline parallelism can be combined, which is also referred to as 3D parallelism. In this paper, we use model parallelism as an umbrella term for both tensor and pipeline parallelism. With an efficient combination of these techniques, we can scale the training of models up to 1 trillion parameters [13].

**Sequence Parallelism**   Sequence parallelism [8] builds on tensor parallelism [19] by further parallelizing normalization and dropout operations along the sequence dimension. This reduces activation memory usage, especially for longer sequences. Efficiently implemented, sequence parallelism does not introduce additional communication overhead when used together with tensor parallelism.

**Activation Checkpointing**   Activation checkpointing [2] enables a tradeoff between memory and compute. Instead of storing all activations for gradients, they are recalculated on the fly during the

| Model | Seq. Len. | GPUs | TP sizes | PP sizes | MB Sizes | Act. Checkpointing | RMSNorm Kernel |
|-------|-----------|------|----------|----------|----------|--------------------|----------------|
| 13B | 2k | 64 | {1, 2} | {1, 2} | {1, 2, 4, 8} | {yes, no} | {yes, no} |
| 13B | 8k | 128 | {1, 2, 4} | {1, 2, 4} | {1, 2, 4} | {yes, no} | {yes, no} |
| 30B | 2k | 256 | {1, 2, 4} | {1, 2, 4} | {1, 2, 4} | {yes, no} | {yes, no} |
| 30B | 8k | 128 | {2, 4} | {2, 4, 8, 16} | {1, 2, 4} | {yes, no} | {yes, no} |
| 65B | 2k | 128 | {2, 4, 8} | {2, 4, 8} | {1, 2, 4} | {yes, no} | {yes, no} |

Table 1: Search space of our training efficiency sweep. We sweep over the Cartesian product of all options given in set notation. In particular, we sweep over different tensor parallelization (**TP**), pipeline parallelization (**PP**), and micro-batch (**MB**) sizes, and also whether activation checkpointing was used. Models with a sequence length of 2k use a global batch size of 2,048, whereas models with a sequence length of 8k use a global batch size of 512. All runs use FLASHATTENTION-2. For runs using activation checkpointing, the RMSNorm kernel caused an error. Therefore, this combination is omitted.

backward pass. This enables fitting larger models into memory and can improve training throughput by enabling larger batch sizes [13].

**Fused Kernels**   Fusing sequential operations into a single kernel enhances the efficiency of memory-bound computations. By executing multiple operations concurrently within a single kernel, data is loaded only once, minimizing memory accesses and optimizing computational overhead.

**Flash Attention**   Dao et al. [5, 4] introduce an IO-aware attention algorithm that builds on kernel fusion. Their method provides speedups compared to a conventional implementation by minimizing read/write operations between the slower high-bandwidth memory and the quicker on-chip SRAM in GPUs. Additionally, selective activation recomputation during the backward pass alleviates the $\mathcal{O}(n^2)$ memory cost in the sequence length.

## 3   Experimental Setup

Our experiments are conducted on up to 32 NVIDIA DGX A100 nodes, each equipped with eight NVIDIA A100 80GB GPUs, resulting in a total of 256 GPUs. The GPUs within each node are interconnected via a third-generation NVLink[1], which provides 600GB/s of bandwidth. Cross-node communication is facilitated by NVIDIA Mellanox 200Gb/s HDR Infiniband[2] connections.

We chose the LLAMA [21] model architecture for our experiments, due to its recent popularity. The LLAMA architecture introduces minor improvements over the standard Transformer architecture [24], which have been incorporated into other models over the past few years. The primary architecture modifications include pre-normalization and RMSNorm [26], the SwiGLU activation function [18], and rotary positional embeddings [20]. Our LLAMA models use a 128k token vocabulary. The LLAMA models have a sequence length of 2k tokens. However, the growing trend of training LLMs with longer sequences [14, 22] led us to assess the training efficiency of our LLAMA models on sequences of up to 8k in length. We use AdamW optimization [11] following the training setup of LLAMA [21]. All training runs are conducted with our in-house large-scale training framework AA-Scaling using mixed-precision with `bfloat16`. We use ZeRO-1 [16] to shard the optimizer states across all data parallel ranks based on the results of previous scaling experiments [13]. The communication framework in use is the `torch.distributed` package with NCCL.

We aim to provide a systematic analysis of different combinations of parallelization strategies and other memory and compute optimizations. To this end, we conducted a large-scale *training efficiency sweep*. We ran this analysis for the following model types: LLAMA 13B (2k & 8k sequence length), LLAMA 30B (2k & 8k sequence length), and LLAMA 65B (2k sequence length). Depending on the model size and availability of compute, we used 64 to 256 GPUs. Table 1 lists the different configuration options for each of the model types. For our training efficiency sweep, we build the Cartesian product of possible options and benchmark each individual configuration. For each configuration, we train for 10 global steps and measure the Model FLOPS Utilization (MFU) [3]. We

---

[1]NVLink: nvidia.com/en-us/data-center/nvlink
[2]Infiniband: nvidia.com/en-us/networking/infiniband-switching

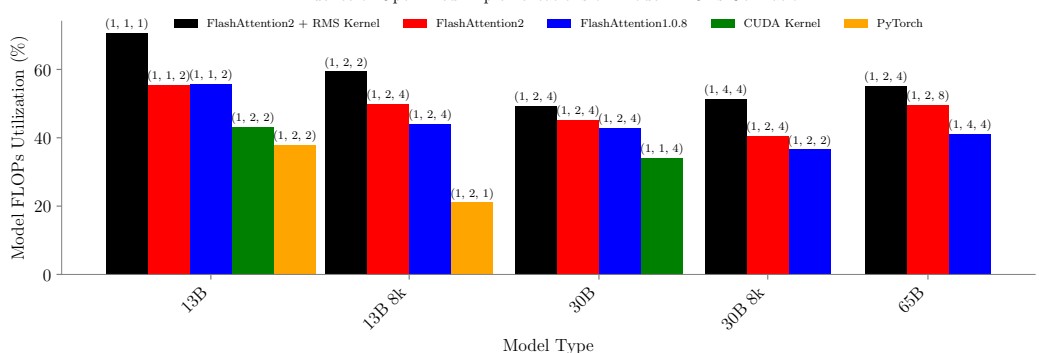

Figure 1: Comparison of the MFU with different attention layer optimizations. The optimal 3D layout was selected for each respective setting. Each optimal layout is annotated with its (`micro-batch size, tensor parallelism size, pipeline parallelism size`). The kernel from Megatron-LM failed to operate with an 8k sequence length.

exclude the first step, as its performance is significantly impacted by a warm-up phase, and report the mean of the last 9 steps.[3] We chose MFU [3] over other metrics such as measured hardware TFLOPS, since the latter are system- and implementation-dependent.

Specifically, we compare different tensor parallelization, pipeline parallelization, and micro-batch sizes, as well as the use of activation checkpointing (yes/no). Since we operate with a fixed number of GPUs and global batch size for each model, the data parallelization size and the number of necessary accumulation steps directly follow from the other specified options and are automatically calculated. For example, using 128 GPUs with a tensor parallelization size of 4 and pipeline parallelization size of 2 results in a rank 16 data parallelization (with $4 \times 2 \times 16 = 128$), each with 2 pipeline stages and each pipeline stage sharded across 4 tensor parallel splits. We provide the full results of our training efficiency sweep in Table B.1.

Additionally, we conducted a preliminary sweep over different attention kernels (native Torch implementation, Megatron-LM kernel[4], FLASHATTENTION-1.0.8, and FLASHATTENTION-2). Based on the results, we concluded that FLASHATTENTION-2 is superior and thus always used it for our main sweep.

In the following section, we will distill the extensive sweep into different, actionable findings that allow us to select the optimal combination of different optimizations.

## 4  Efficient LLM Training Analysis

### 4.1  Fused Kernels and Flash Attention

Our evaluation of FLASHATTENTION expands on the evaluations present in the original papers [5, 4]. While those studies compared the efficiency of FLASHATTENTION for models up to 2.7B parameters on a single node, we scaled our experiments to substantially larger model sizes and also up to 256 GPUs. We further compare with a more optimized baseline, the Megatron-LM softmax attention kernel. Additionally, we evaluate the use of an optimized RMSNorm kernel from the FLASHATTENTION repository.

#### 4.1.1  Attention

In Figure 1, we present results from both our main and preliminary sweeps over attention implementations, detailed in Section 3. We compare the following different kernels: FLASHATTENTION-2, FLASHATTENTION-1.0.8, the Megatron-LM kernel, and the standard PyTorch implementation. The fused kernel does not support a sequence length exceeding 2,048 tokens. Due to the underperformance

---

[3]Our detailed MFU calculation is reported in Appendix A.1.
[4]Megatron-LM softmax attention kernel: from here (Date: 28 July 2022)

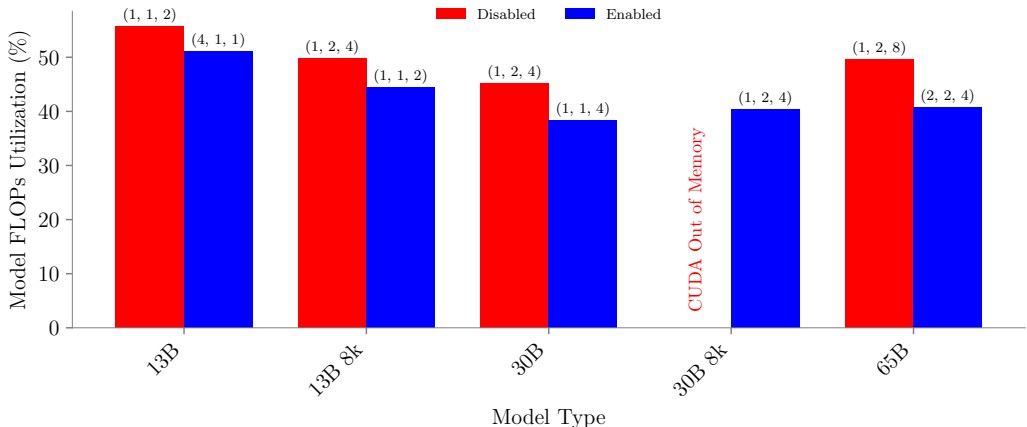

Figure 2: Comparing MFU of the optimal 3D layout with and without activation checkpointing. LLAMA 30B with 8k sequence length did not fit into memory without checkpointing. The reported results do not use the RMSNorm kernel. Each optimal layout is annotated with its (`micro-batch size`, `tensor parallelism size`, `pipeline parallelism size`).

of pure PyTorch attention in our LLAMA 13B evaluation, we did not include it for larger models. For LLAMA 65B and 30B with 8k sequence length, we only considered FLASHATTENTION.

Unsurprisingly, we find that FLASHATTENTION vastly outperforms the native PyTorch implementation. However, we also find that FLASHATTENTION significantly outperforms the kernel from Megatron-LM. Between the two different FLASHATTENTION versions, FLASHATTENTION-2 outperforms FLASHATTENTION-1.0.8 by 4 to 13 percentage points across model sizes. FLASHATTENTION-1.0.8 already contains many of the optimizations introduced in the FLASHATTENTION-2 paper, which measures a $2\times$ improvement [4].

It is important to note that FLASHATTENTION's improvements are two-fold: FLASHATTENTION's improved tiling method for an efficient IO-aware SRAM cache utilization and reduced memory requirements through its activation recomputation approach in the attention block. Notably, all best-performing FLASHATTENTION layouts reported in Figure 1 do not make use of activation checkpointing, thereby also benefiting from FLASHATTENTION's own activation recomputation.

### 4.1.2 RMSNorm Kernel

We also evaluate the effect of FLASHATTENTION's optimized RMSNorm kernel in Figure 1. We see that the RMSNorm kernel provides a significant boost in training efficiency, up to 14 percentage points compared to FLASHATTENTION-2 without the RMSNorm kernel. Notably, with the use of the kernel, we can fit the entire LLAMA 13B model into a single GPU without model parallelization during training (although we still employ ZeRO-1 and shard the optimizer states). In general, the RMSNorm kernel allows us to choose more efficient parallelization layouts due to its memory savings. We do not have results combining activation checkpointing with the RMSNorm kernel, as the combination caused an error in our experiments. We control for this and only consider runs without the RMSNorm kernel whenever necessary for a fair comparison.

### 4.2 Activation Checkpointing

In Figure 2, we report the MFU of the best configurations across model sizes, both with activation checkpointing of every layer and without. Overall, we see that not using activation checkpointing and compensating for the incurred memory cost with smaller batch sizes or a higher degree of model parallelism achieves the best training throughputs. For a fair comparison, we do not include runs with the RMSNorm kernel, since the kernel caused an error when coupled with checkpointing.

For the LLAMA 30B with 8k sequence length, activation checkpointing was necessary to fit the model into memory during training, even with tensor parallelism sizes up to 4 and pipeline parallelism

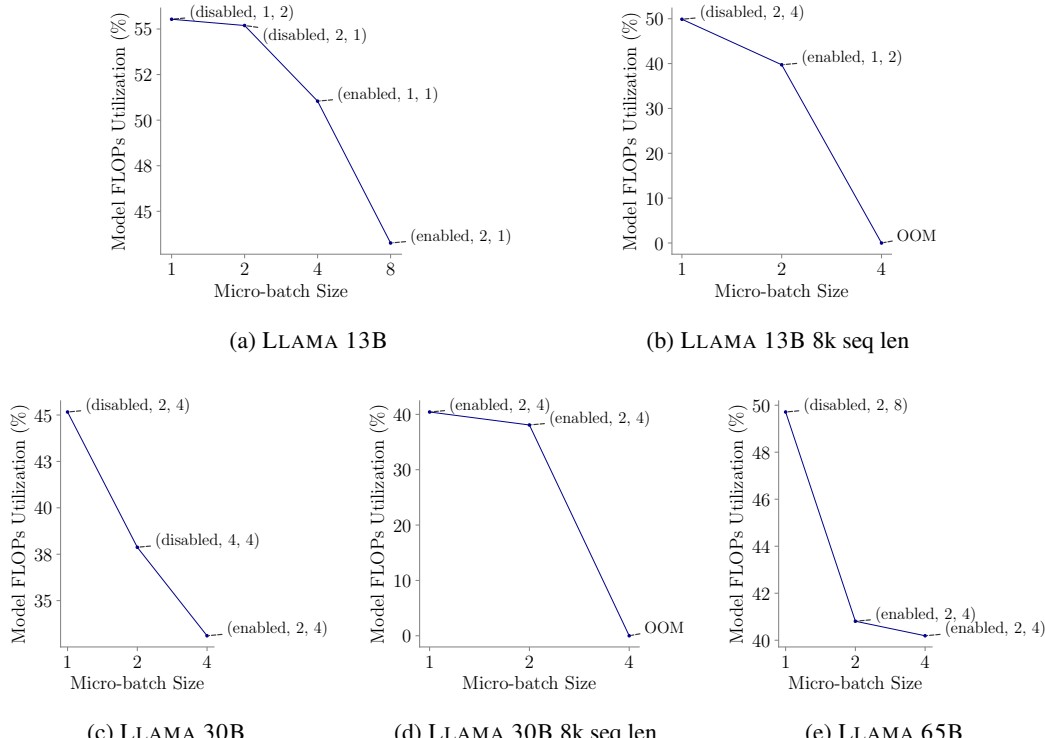

Figure 3: MFU of the best-performing run configurations at different fixed micro-batch sizes, visualized by the (`activation checkpointing`, `tensor parallelism size`, `pipeline parallelism size`) triple. The reported results do not use the RMSNorm kernel.

sizes of up to 16. We could not increase the tensor parallelism because the LLAMA 30B model has 52 attention heads, which are not divisible by 8. In Section 4.1, we show that adding the RMSNorm kernel further reduces the required memory so that activation checkpointing does not become necessary. In this case, we again see that a layout without activation checkpointing achieves the best throughput.[5]

It is crucial to underline that achieving efficient performance for such model sizes without activation checkpointing is only feasible due to FLASHATTENTION. Without FLASHATTENTION, any runs exceeding the size of LLAMA 13B required the use of activation checkpointing due to out-of-memory errors, despite the high degrees of parallelization we considered as part of our sweep.

FLASHATTENTION already employs its own selective activation checkpointing in the attention block. These findings suggest activation checkpointing for large-scale Transformer training needs to be more targeted. Previous work [8] has also questioned the need for checkpointing in every layer and suggests a selective activation recomputation approach within the attention block. However, with the introduction of FLASHATTENTION, the focus on selective activation recomputation within the attention block arguably becomes less important, as this is already covered in an efficient manner within FLASHATTENTION. Nevertheless, a promising approach can be the application of selective activation recomputation only to the MLP block, thereby complementing FLASHATTENTION's inherent activation recomputation of the attention block. Recently, this issue was tackled with an activation checkpointing strategy that is aware of FLASHATTENTION's activation recomputation [9].

### 4.3 Micro-batch size

In this section, we evaluate the tradeoff between the micro-batch size and required degree of model (tensor or pipeline) parallelism and activation checkpointing. Previous work [13] benchmarked different micro-batch sizes with fixed degrees of tensor and pipeline parallelism and show that larger micro-batch sizes lead to higher throughput. However, a smaller micro-batch size might enable

---

[5]Detailed results are reported in Appendix B.5.

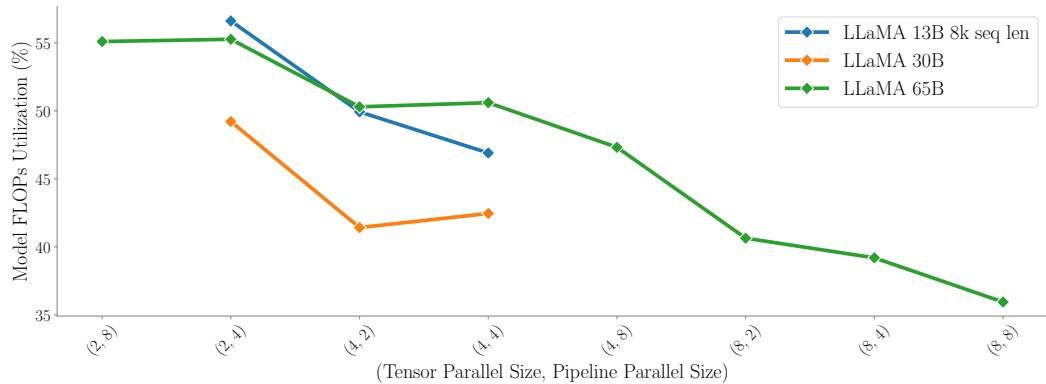

Figure 4: MFU for various model and pipeline parallel configurations for the LLAMA 13B with 8k sequence length, LLAMA 30B, and LLAMA 65B models. Only runs with a micro-batch size of 1, activation checkpointing disabled, FLASHATTENTION-2, and the RMS norm kernel are included; runs that ran out of memory are excluded. The LLAMA 13B and the LLAMA 30B with 8k sequence length models are excluded due to limited model parallel configuration options in our sweep.

a different, more efficient parallelization configuration. Also, the degree of tensor and pipeline parallelism are often not fixed in practice.

In Figure 3, we show the best performing (`activation checkpointing, tensor parallelism size, pipeline parallelism size`) configuration for each of our assessed model types. To fairly evaluate activation checkpointing, we do not include runs with the RMSNorm kernel, since the kernel resulted in an error when coupled with checkpointing. We see that for all model types, a micro-batch size of 1 achieves the best MFU. In general, we find: the smaller the micro-batch size, the better the MFU. The models with an 8k sequence length did not fit into memory with any configuration when using a micro-batch size bigger than 2.

Thus, we conclude that choosing a micro-batch size of 1 is beneficial in most scenarios. The superior performance of a micro-batch size of 1 can be attributed to the following three factors.

**Minimal Degree of Model Parallelization:** The most efficient training typically requires the least amount of model (tensor or pipeline) parallelization, which is achieved when the micro-batch size is smallest.

**Avoiding Activation Checkpointing:** For some models (e.g., LLAMA 65B), a micro-batch size of 1 was the only configuration allowing training without activation check-pointing. As discussed in the previous section, not using activation checkpointing often allows for the highest throughput configurations. The LLAMA 30B 8k model did not fit into memory without using the RMSNorm kernel.

**Reduced Pipeline Bubble Time:** A smaller micro-batch size reduces the time spent in the pipeline bubble at the beginning and end of a batch. We already use the better-than-naive Pipedream 1F1B scheduling method [12] discussed in Section 2.

### 4.4 Tensor vs. Pipeline Parallelism

Narayanan et al.'s [13] ablation studies show that neither tensor nor pipeline parallelism, when used in isolation, can achieve the performance of utilizing both at the same time. Our empirical data, especially from the LLAMA 65B model – where higher degrees of parallelism are necessary – validate this to some extent even in combination with the newly introduced optimizations, as depicted in Figure 4. Their results suggest that an even distribution between the tensor and pipeline parallelism size is optimal, up until the tensor parallelism size reaches the GPU limit in a single node. In contrast, our results favor pipeline parallelism over tensor parallelism. The LLAMA 65B model performed best with a (tensor, pipeline) parallelism size of $(2, 8)$ compared to an evenly distributed $(4, 4)$. Also, the $(8, 2)$ configuration was considerably less efficient. This trend was also observed in the LLAMA 13B with 8k sequence length and LLAMA 30B model, where the configurations with a

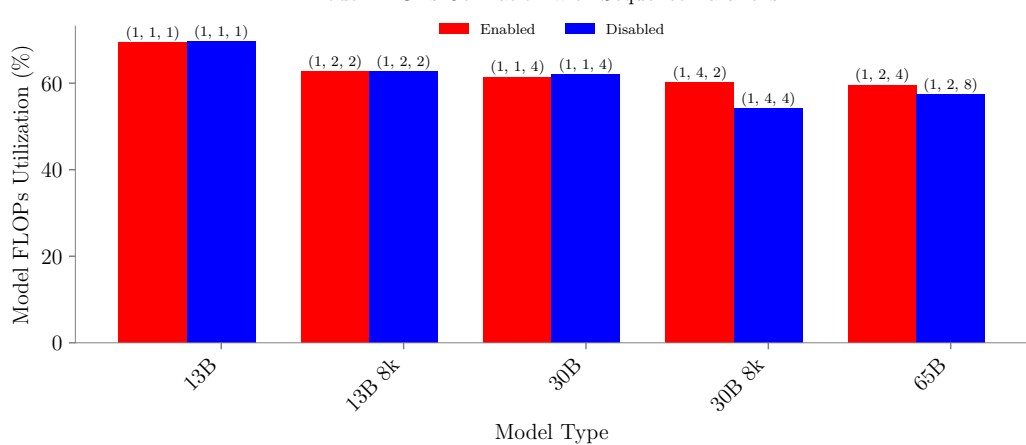

Figure 5: Comparing MFU of the optimal 3D layout with and without sequence parallelism. The reported results use the RMSNorm kernel. Each optimal layout is annotated with its (`micro-batch size, tensor parallelism size, pipeline parallelism size`).

higher pipeline parallel size outperform the configurations with a larger tensor parallelism size. The training efficiency measured by Megatron-LM [19] was comparable when the tensor and pipeline parallelism sizes were interchanged.

### 4.5 Sequence Parallelism

In this section, we perform an additional efficiency sweep to assess the impact of sequence parallelism. Based on the findings from previous sections, we limited the search space to consistently use FLASHATTENTION-2 and the RMSNorm kernel, while omitting the use of activation checkpointing. Furthermore, we restricted the number of GPUs for each model type due to computational constraints. The full configuration sweep for each model type is documented in Table 9.

In Figure 5, we report the MFU of the best configurations across model sizes, both with sequence parallelism enabled or disabled. For the LLAMA 13B and 30B models with a 2k sequence length, the top configurations do not employ any tensor parallelism, hence the activation of sequence parallelism shows no effect. In the case of the 13B model with an 8k sequence length, the top configuration employs a tensor parallelism size of 2; however, no improvement in training efficiency is observed. For the largest models and sequence lengths, the 30B with 8k sequence length and the 65B models, we can see 2-6 percentage point improvements when using sequence parallelism. In both cases, sequence parallelism enables a lower degree of model parallelization due to the reduced memory requirements, leading to higher training efficiency.

Therefore, we conclude that the use of sequence parallelism, when paired with several other optimizations explored in this work, only facilitates a notable difference in training efficiency for model sizes exceeding 30B parameters or 2k sequence length.

### 4.6 End-to-End Performance

We evaluate the recommendations distilled from our extensive training efficiency sweep against other publicly reported results in Table 2. For our runs using the in-house AA-Scaling framework, we report the MFU of the best configuration for each model size, following our recommendations.

We evaluate against publicly available benchmarks from MosaicML[6], Megatron-DeepSpeed [8], Meta's LLAMA [21], and Megatron-LM [13]. Other frameworks were excluded from our comparisons, because they either lacked publicly available training efficiency scores, used entirely different hardware, or trained models with vastly different parameter sizes. To the best of our knowledge, we

---

[6]MosaicML MPT Training Benchmarks: from here (Date: 26 September 2023)

| Model | GPUs | Seq. Len. | Batch Size | MFU (↑) |
|---|---|---|---|---|
| **AA-Scaling LLAMA 13B (ours)** | 64 | 2048 | 2048 | **70.5%** |
| MPT 13B | 64 | 2048 | 2048 | 52.5% |
| Megatron-LM 18B[†] | 256 | 2048 | 1024 | 34.2% |
| **AA-Scaling LLAMA 13B (ours)** | 64 | 8192 | 512 | **62.7%** |
| MPT 13B | 8 | 8192 | 120 | 52.8% |
| **AA-Scaling LLAMA 30B (ours)** | 64 | 2048 | 2048 | **61.9%** |
| MPT 30B | 64 | 2048 | 3072 | 52.9% |
| Megatron-DeepSpeed 22B | 8 | 2048 | 4 | 41.5% |
| Megatron-LM 39B[†] | 512 | 2048 | 1536 | 34.5% |
| **AA-Scaling LLAMA 30B (ours)** | 64 | 8192 | 512 | **60.2%** |
| MPT 30B | 8 | 8192 | 168 | 42.6% |
| **AA-Scaling LLAMA 65B (ours)** | 64 | 2048 | 2048 | **59.6%** |
| MPT 70B | 64 | 2048 | 2048 | 53.3% |
| LLAMA 65B by Meta[†] | 2048 | 2048 | 2048 | 49.4% |
| Megatron-LM 76B[†] | 1024 | 2048 | 1792 | 34.7% |

Table 2: Best achieved end-to-end training efficiency numbers using our recommendations compared to other public training efficiency numbers. We group across comparable model sizes and sequence length. Batch size refers to the *Global* Batch size. [†]: MFU numbers were calculated by us based on published training times, as detailed in Appendix A. We provide the exact configurations of our runs included in this table in Appendix B.1.

have gathered the best performing, publicly available training efficiency benchmarks for LLMs with comparable architectures and parameter counts.

In general, our best configurations achieve the highest MFU numbers, setting the state-of-the-art for all our assessed model sizes. The improvements on top of the previous state-of-the-art range from 6–18 MFU percentage points. Noticeably our LLAMA 13B model achieves an MFU of 70.5%, outperforming the MPT and Megratron-LM models. For the 13B and 30B models with an 8k sequence length, our only point of comparison are the models by MosaicML's framework. Here, we outperform the MPT models by 9–17 percentage points. Within the 65B parameter model range, we outperform MPT-70B, the original LLAMA 65B model by Meta, and the Megatron-LM 76B model with an MFU of 59.6% compared to 53.3%, 49.4%, and 34.7%, respectively.

We hope that our findings can contribute to pushing the envelope of efficiently utilizing hardware accelerators for large-scale model training.

**Note on comparability.** Most of the comparisons in Table 2 are not one-to-one, due to further differences such as model architecture, employed global batch size, number of used GPUs, and hardware interconnect. For example, the MPT model family employs additional efficiency optimizations, such as the use of ALiBi [15] instead of RoPE [20]. On the other hand, our models use a 128k token vocabulary, which can result in a more optimistic MFU estimate compared to smaller vocabulary sizes. The comparison made in Table 2 is not meant to be directly one-to-one, but a wholesale evaluation of the achieved end-to-end training efficiency. We hope to showcase that a careful evaluation of the training layout and optimizations used can enable a significant boost to training efficiency.

## 5 Conclusion

We conducted an exhaustive search over possible combinations of tensor, pipeline, and data parallelism layouts, fused kernels, FLASHATTENTION, activation checkpointing, micro-batch sizes, and sequence parallelism. Based on our findings, we make the following recommendations:

- Use a micro-batch size of 1 to enable the least degree of model parallelization, to avoid activation checkpointing, and to reduce pipeline bubbles.

- Prefer increasing the degree of tensor and pipeline parallelization over the use of activation checkpointing.
- Only scale the micro-batch size when you cannot further reduce the degree of model parallelization.
- Use sequence parallelization for models exceeding 30B parameters and 2k sequence length.

We experimentally verify that the efficacy of the FLASHATTENTION-2 kernel remains as we scale the model size to tens of billions of parameters and to training on multiple nodes. Additionally, we compared the end-to-end training efficiency of our most efficient configurations against several other frameworks. We achieved state-of-the-art efficiency in five out of the five model configurations we evaluated, reaching up to 70.5% MFU.

For future work, reconciling activation checkpointing with FLASHATTENTION via a more selective approach that targets the MLP block presents an exciting opportunity. Also, NVIDIA's recently released H100 GPUs with more efficient support for `fp8` precision might enable new training strategies, which should be evaluated.

We publish the full data of our sweeps on GitHub at `https://github.com/Aleph-Alpha/ NeurIPS-WANT-submission-efficient-parallelization-layouts`.

## Limitations

**Applicability of recommendations to other frameworks.** We expect that our recommendation results will be applicable to frameworks, such as Megatron-DeepSpeed, that utilize similar 3D parallel training configurations and ZeRO-1. However, some recommendations may be less relevant to frameworks that use different ZeRO stages [16], FSDP [27], or other parallelization strategies [7].

**Global batch size and number of GPUs considerations.** The global batch size for our experiments was set based on the pre-training of the original LLAMA. Reducing the global batch size or increasing the number of GPUs used during training might lead to a lower model FLOPs utilization than reported in our experiments. This can be attributed to fewer gradient accumulation steps, which could increase the pipeline bubble's share.

**Applicability to other Model Architectures and Domains.** We perform our training analysis using a Transformer language model with the LLAMA architecture. Some of the benchmarked optimizations are specific to the general Transformer architecture, such as FLASHATTENTION, or to specific choices in architecture design, such as the use of RMSNorm. In our experiments, we only considered the language modeling task. Applications of the Transformer architecture to other domains, such as vision [6], might also benefit from our recommendations. However, we did not experimentally verify this.

**Applicability to other Hardware Accelerators.** The scope of our training analysis was limited to NVIDIA DGX A100 nodes, connected via high-speed Infiniband. The use of slower interconnects or other hardware accelerators, such as TPUs or AMD GPUs, might affect the applicability of our recommendations.

**End-to-End Performance Comparison.** We stress again that the comparison of training efficiency in Table 2 is not suitable as a one-to-one comparison due to differences in model architecture, employed hardware, and training settings.

## Acknowledgements

We thank the anonymous reviewers for their helpful comments as well as the research team within the lab at Aleph Alpha for insightful discussions and feedback. We acknowledge the financial support for one of the authors by the German Federal Ministry for Education and Research (BMBF) through the project «KI-Servicezentrum Berlin Brandenburg» (01IS22092).

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

# A    Measuring Model Training Efficiency Calculations

## A.1    Model FLOPs Utilization (MFU)

The calculation of Model FLOPs Utilization (MFU) follows that of PaLM [3]. We consider the theoretical peak matrix multiplication throughput of $P$ FLOPs per second (e.g., A100 GPUs with 312 peak matmul TFLOPs). Then, the model FLOPs utilization is the ratio of the achieved throughput in tokens per second to the theoretical peak throughput $R = P/(6N + 12LHQT)$, where $L$ is the number of layers, $H$ the number of attention heads, $Q$ the attention head size, and $T$ the sequence length. Note that $L \times H$ is equal to the hidden layer size of the model.

```python
class HardwareType(Enum):
    A100 = "a100"
    H100 = "h100"
    RTX3090 = "rtx3090"

    @property
    def max_tflops(self):
        """
        Mappings for Maximum Throughput numbers of each GPU.
        Only includes FP16 for now.
        """
        max_tflop_mapping = {"a100": 312e12, "h100": 989.4e12, "rtx3090": 35.58e12}
        return max_tflop_mapping[self.value]

def get_model_flop_utilizations_palm(
    iter_time_s: float,
    parameter_count: int,
    topology: Any,
    architecture: Any,
    hardware: HardwareType = HardwareType.A100,
):
    tokens_per_second = (
        topology.config.global_batch_size * architecture.sequence_length
    ) / iter_time_s
    theoretical_peak_matmul_throughput = (
        hardware.max_tflops * topology.config.world_size
    )
    attention_flops = (
        12
        * architecture.num_layers
        * architecture.hidden_size
        * architecture.sequence_length
    )
    model_flops = 6 * parameter_count + attention_flops
    theoretical_peak_throughput = theoretical_peak_matmul_throughput / model_flops
    model_flops_utilization = tokens_per_second / theoretical_peak_throughput
    return model_flops_utilization
```

## A.2  LLAMA MFU

From LLAMA paper [21]:

> "When training a 65B-parameter model, our code processes around 380 tokens/sec/GPU on 2048 A100 GPU with 80GB of RAM. This means that training over our dataset containing 1.4T tokens takes approximately 21 days."

LLAMA 65B model FLOPs utilization (MFU):

```python
tokens_per_second = 380 * 2048
theoretical_peak_matmul_throughput = (
    312e12 * 2048
)
attention_flops = (
    12
    * 80
    * 8192
    * 2048
)
model_flops = 6 * 65e9 + attention_flops
theoretical_peak_throughput = theoretical_peak_matmul_throughput / model_flops
model_flops_utilization = tokens_per_second / theoretical_peak_throughput
```

LLAMA 65B MFU = 49.46%

## A.3  Megatron-LM MFU

Based on Megatron-LM's [13] provided formula, the end-to-end training time is given by $\approx \frac{8TP}{nX}$, where $T$ represents the number of tokens, $P$ the number of parameters, $n$ the number of GPUs, and $X$ the achieved TFLOPs per GPU. Since they provide the achieved TFLOPs per GPU, we can determine the step time through the formula. Subsequently, we can compute the MFU using the specified architecture configuration of the model.

**Megatron-LM 18B:**

Step time $= \frac{8 \cdot 1024 \cdot 2048 \cdot 18.4 \cdot 10^9}{256 \cdot 135 \cdot 10^{12}} s = 8.93s$

```
tokens_per_second = (1024 * 2048) / 8.93
theoretical_peak_matmul_throughput = (
    312e12 * 256
)
attention_flops = (
    12
    * 40
    * 6144
    * 2048
)
model_flops = 6 * 18.4e9 + attention_flops
theoretical_peak_throughput = theoretical_peak_matmul_throughput / model_flops
model_flops_utilization = tokens_per_second / theoretical_peak_throughput
```

Megatron-LM 18B MFU = 34.24%

**Megatron-LM 39B:**

Step time $= \frac{8 \cdot 1536 \cdot 2048 \cdot 39.1 \cdot 10^9}{512 \cdot 138 \cdot 10^{12}} s = 13.92s$

```
tokens_per_second = (1536 * 2048) / 13.92
theoretical_peak_matmul_throughput = (
    312e12 * 512
)
attention_flops = (
    12
    * 48
    * 8192
    * 2048
)
model_flops = 6 * 39.1e9 + attention_flops
theoretical_peak_throughput = theoretical_peak_matmul_throughput / model_flops
model_flops_utilization = tokens_per_second / theoretical_peak_throughput
```

Megatron-LM 39B MFU = 34.56%

**Megatron-LM 76B:**

Step time $= \frac{8 \cdot 1792 \cdot 2048 \cdot 76.1 \cdot 10^9}{1024 \cdot 140 \cdot 10^{12}} s = 15.59s$

```
tokens_per_second = (1792 * 2048) / 15.59
theoretical_peak_matmul_throughput = (
    312e12 * 1024
)
attention_flops = (
    12
    * 60
    * 10240
    * 2048
)
model_flops = 6 * 76.1e9 + attention_flops
theoretical_peak_throughput = theoretical_peak_matmul_throughput / model_flops
model_flops_utilization = tokens_per_second / theoretical_peak_throughput
```

Megatron-LM 76B MFU = 34.76%

# B   Training Efficiency Sweeps

## B.1   End-to-End Best-Performing Runs

Table 3: Configurations for the end-to-end performance results of Table 2. All best-performing layouts use FLASHATTENTION-2, the RMS norm kernel, and do not make use of activation checkpointing.

| Model | GPUs | Step Time | MFU | MB Size | TP size | PP Size | Sequence Parallel |
|---|---|---|---|---|---|---|---|
| **AA-Scaling LLAMA 13B** | 64 | 3.32 | 70.57 | 1 | 1 | 1 | False |
| **AA-Scaling LLAMA 13B 8k seq len** | 64 | 34.84 | 62.78 | 1 | 1 | 2 | True |
| **AA-Scaling LLAMA 30B** | 64 | 72.00 | 61.98 | 1 | 1 | 4 | False |
| **AA-Scaling LLAMA 30B 8k seq len** | 64 | 84.37 | 60.22 | 1 | 4 | 2 | True |
| **AA-Scaling LLAMA 65B** | 64 | 147.02 | 59.62 | 1 | 2 | 4 | True |

## B.2   LLAMA 13B

Table 4: Performance analysis of a LLAMA 13B model trained on 64 A100 GPUs with the AA-Scaling codebase: A comprehensive sweep of 3D parallel layout configurations. The table displays variations in step time and MFU across different activation checkpointing types, kernels, micro batch (MB) sizes, tensor parallel (TP) sizes, and pipeline parallel (PP) sizes. The analysis also includes Out of Memory (OOM) error occurrences.

| Step Time | MFU | Activation | Kernel | MB Size | TP Size | PP Size |
|---|---|---|---|---|---|---|
| 26.54 | 70.57 | disabled | flash_attn2 + RMS kern. | 1 | 1 | 1 |
| 29.70 | 63.05 | disabled | flash_attn2 + RMS kern. | 2 | 2 | 1 |
| 31.06 | 60.26 | disabled | flash_attn2 + RMS kern. | 1 | 1 | 2 |
| 31.29 | 59.82 | disabled | flash_attn2 + RMS kern. | 1 | 2 | 1 |
| 33.11 | 56.55 | disabled | flash_attn2 + RMS kern. | 2 | 1 | 2 |
| 33.58 | 55.71 | disabled | flash_attn1.0.8 | 1 | 1 | 2 |
| 33.70 | 55.53 | disabled | flash_attn2 | 1 | 1 | 2 |
| 33.90 | 55.19 | disabled | flash_attn2 | 2 | 2 | 1 |
| 34.84 | 53.69 | disabled | flash_attn2 + RMS kern. | 1 | 2 | 2 |
| 35.05 | 53.37 | disabled | flash_attn2 + RMS kern. | 2 | 2 | 2 |
| 35.54 | 52.64 | disabled | flash_attn2 | 1 | 2 | 1 |
| 36.66 | 51.04 | every_layer | flash_attn2 | 4 | 1 | 1 |
| 36.69 | 51.02 | every_layer | flash_attn2 | 2 | 1 | 1 |
| 37.51 | 49.89 | every_layer | flash_attn2 | 1 | 1 | 1 |
| 37.57 | 49.80 | disabled | flash_attn2 + RMS kern. | 4 | 2 | 2 |
| 37.71 | 49.59 | disabled | flash_attn1.0.8 | 2 | 2 | 2 |
| 38.27 | 48.86 | disabled | flash_attn2 | 2 | 1 | 2 |
| 39.64 | 47.19 | disabled | flash_attn2 | 1 | 2 | 2 |
| 39.83 | 46.97 | disabled | flash_attn2 | 2 | 2 | 2 |
| 40.25 | 46.46 | disabled | flash_attn1.0.8 | 1 | 2 | 2 |
| 40.56 | 46.11 | disabled | flash_attn1.0.8 | 1 | 2 | 1 |
| 41.81 | 44.74 | disabled | flash_attn1.0.8 | 2 | 2 | 1 |
| 42.30 | 44.21 | every_layer | flash_attn1.0.8 | 2 | 1 | 2 |
| 42.36 | 44.14 | every_layer | flash_attn1.0.8 | 1 | 1 | 2 |
| 42.54 | 43.96 | every_layer | flash_attn1.0.8 | 4 | 1 | 2 |
| 43.13 | 43.36 | every_layer | flash_attn1.0.8 | 4 | 1 | 1 |
| 43.24 | 43.26 | every_layer | flash_attn2 | 8 | 2 | 1 |
| 43.36 | 43.13 | disabled | fused | 1 | 2 | 2 |
| 43.89 | 42.61 | every_layer | flash_attn1.0.8 | 2 | 1 | 1 |
| 44.10 | 42.40 | every_layer | flash_attn2 | 4 | 2 | 1 |
| 44.13 | 42.39 | every_layer | flash_attn2 | 1 | 1 | 2 |
| 44.35 | 42.18 | every_layer | flash_attn1.0.8 | 1 | 1 | 1 |
| 44.45 | 42.09 | every_layer | flash_attn2 | 2 | 2 | 1 |
| 44.72 | 41.82 | every_layer | flash_attn1.0.8 | 8 | 1 | 2 |
| 45.96 | 40.69 | disabled | fused | 1 | 2 | 1 |
| 46.86 | 39.90 | every_layer | fused | 2 | 1 | 2 |
| 47.12 | 39.72 | every_layer | flash_attn2 | 1 | 2 | 1 |
| 47.09 | 39.71 | every_layer | flash_attn1.0.8 | 4 | 2 | 1 |
| 47.17 | 39.64 | every_layer | fused | 4 | 1 | 2 |
| 47.37 | 39.51 | every_layer | flash_attn2 | 2 | 1 | 2 |
| 47.50 | 39.36 | every_layer | flash_attn1.0.8 | 2 | 2 | 1 |
| 47.82 | 39.11 | every_layer | fused | 4 | 1 | 1 |
| 47.95 | 39.00 | every_layer | fused | 1 | 1 | 2 |
| 48.93 | 38.23 | every_layer | fused | 1 | 1 | 1 |
| 49.35 | 37.89 | disabled | torch | 1 | 2 | 2 |
| 49.39 | 37.87 | every_layer | fused | 2 | 1 | 1 |
| 49.77 | 37.57 | every_layer | flash_attn1.0.8 | 4 | 2 | 2 |

| Step Time | MFU | Activation | Kernel | MB Size | TP size | PP Size |
|---|---|---|---|---|---|---|
| 50.05 | 37.37 | every_layer | flash_attn1.0.8 | 1 | 2 | 1 |
| 50.16 | 37.28 | every_layer | flash_attn1.0.8 | 8 | 2 | 2 |
| 51.44 | 36.41 | every_layer | flash_attn1.0.8 | 2 | 2 | 2 |
| 51.79 | 36.11 | every_layer | fused | 4 | 2 | 1 |
| 52.55 | 35.59 | disabled | torch | 1 | 2 | 1 |
| 53.50 | 34.95 | every_layer | flash_attn1.0.8 | 1 | 2 | 2 |
| 53.57 | 34.94 | every_layer | flash_attn2 | 2 | 2 | 2 |
| 54.17 | 34.52 | every_layer | torch | 1 | 1 | 2 |
| 54.35 | 34.40 | every_layer | fused | 2 | 2 | 1 |
| 54.56 | 34.27 | every_layer | fused | 2 | 2 | 2 |
| 55.13 | 33.95 | every_layer | fused | 4 | 2 | 2 |
| 55.35 | 33.78 | every_layer | torch | 4 | 1 | 2 |
| 55.61 | 33.64 | every_layer | flash_attn2 | 4 | 1 | 2 |
| 56.21 | 33.41 | every_layer | torch | 2 | 1 | 2 |
| 55.99 | 33.40 | every_layer | torch | 1 | 1 | 1 |
| 56.61 | 33.06 | every_layer | flash_attn2 | 4 | 2 | 2 |
| 57.07 | 32.78 | every_layer | torch | 2 | 1 | 1 |
| 57.25 | 32.66 | every_layer | fused | 1 | 2 | 1 |
| 58.92 | 31.77 | every_layer | fused | 1 | 2 | 2 |
| 60.24 | 31.04 | every_layer | torch | 4 | 2 | 1 |
| 60.75 | 30.78 | every_layer | torch | 2 | 2 | 1 |
| 62.60 | 29.87 | every_layer | torch | 1 | 2 | 1 |
| 62.69 | 29.83 | every_layer | torch | 2 | 2 | 2 |
| 63.28 | 29.58 | every_layer | torch | 4 | 2 | 2 |
| 65.25 | 28.66 | every_layer | torch | 1 | 2 | 2 |
| 65.47 | 28.57 | every_layer | flash_attn2 | 8 | 2 | 2 |
| 72.82 | 25.69 | every_layer | flash_attn2 | 8 | 1 | 2 |
| OOM Error | | disabled | flash_attn2 + RMS kern. | 8 | 2 | 1 |
| OOM Error | | disabled | flash_attn2 + RMS kern. | 8 | 1 | 2 |
| OOM Error | | disabled | flash_attn2 + RMS kern. | 4 | 1 | 2 |
| OOM Error | | disabled | flash_attn2 + RMS kern. | 2 | 1 | 1 |
| OOM Error | | disabled | flash_attn2 + RMS kern. | 4 | 1 | 1 |
| OOM Error | | disabled | flash_attn2 + RMS kern. | 4 | 2 | 1 |
| OOM Error | | disabled | flash_attn2 + RMS kern. | 8 | 1 | 1 |
| OOM Error | | disabled | flash_attn2 | 8 | 2 | 2 |
| OOM Error | | disabled | flash_attn2 | 4 | 2 | 1 |
| OOM Error | | disabled | flash_attn2 | 8 | 1 | 2 |
| OOM Error | | disabled | flash_attn2 | 4 | 1 | 2 |
| OOM Error | | disabled | flash_attn2 | 4 | 1 | 1 |
| OOM Error | | disabled | flash_attn2 | 8 | 2 | 1 |
| OOM Error | | disabled | flash_attn2 | 2 | 1 | 1 |
| OOM Error | | disabled | flash_attn2 | 1 | 1 | 1 |
| OOM Error | | disabled | flash_attn2 | 4 | 2 | 2 |
| OOM Error | | every_layer | torch | 8 | 2 | 2 |
| OOM Error | | disabled | torch | 8 | 2 | 2 |
| OOM Error | | every_layer | fused | 8 | 2 | 2 |
| OOM Error | | disabled | fused | 8 | 2 | 2 |
| OOM Error | | disabled | flash_attn1.0.8 | 8 | 2 | 2 |
| OOM Error | | disabled | torch | 4 | 2 | 2 |
| OOM Error | | disabled | fused | 4 | 2 | 2 |
| OOM Error | | disabled | flash_attn1.0.8 | 4 | 2 | 2 |
| OOM Error | | disabled | torch | 2 | 2 | 2 |
| OOM Error | | disabled | fused | 2 | 2 | 2 |
| OOM Error | | every_layer | torch | 8 | 1 | 2 |
| OOM Error | | disabled | torch | 8 | 1 | 2 |
| OOM Error | | every_layer | fused | 8 | 1 | 2 |
| OOM Error | | disabled | fused | 8 | 1 | 2 |
| OOM Error | | disabled | flash_attn1.0.8 | 8 | 1 | 2 |
| OOM Error | | disabled | torch | 4 | 1 | 2 |
| OOM Error | | disabled | fused | 4 | 1 | 2 |
| OOM Error | | disabled | flash_attn1.0.8 | 4 | 1 | 2 |
| OOM Error | | disabled | torch | 2 | 1 | 2 |
| OOM Error | | disabled | fused | 2 | 1 | 2 |
| OOM Error | | disabled | flash_attn1.0.8 | 2 | 1 | 2 |
| OOM Error | | disabled | torch | 1 | 1 | 2 |
| OOM Error | | disabled | fused | 1 | 1 | 2 |
| OOM Error | | every_layer | torch | 8 | 2 | 1 |
| OOM Error | | disabled | torch | 8 | 2 | 1 |
| OOM Error | | every_layer | fused | 8 | 2 | 1 |
| OOM Error | | disabled | fused | 8 | 2 | 1 |
| OOM Error | | every_layer | flash_attn1.0.8 | 8 | 2 | 1 |
| OOM Error | | disabled | flash_attn1.0.8 | 8 | 2 | 1 |
| OOM Error | | disabled | torch | 4 | 2 | 1 |
| OOM Error | | disabled | fused | 4 | 2 | 1 |
| OOM Error | | disabled | flash_attn1.0.8 | 4 | 2 | 1 |
| OOM Error | | disabled | torch | 2 | 2 | 1 |
| OOM Error | | disabled | fused | 2 | 2 | 1 |
| OOM Error | | every_layer | torch | 8 | 1 | 1 |
| OOM Error | | disabled | torch | 8 | 1 | 1 |

| Step Time | MFU | Activation | Kernel | MB Size | TP size | PP Size |
|---|---|---|---|---|---|---|
| OOM Error | | every_layer | fused | 8 | 1 | 1 |
| OOM Error | | disabled | fused | 8 | 1 | 1 |
| OOM Error | | every_layer | flash_attn1.0.8 | 8 | 1 | 1 |
| OOM Error | | disabled | flash_attn1.0.8 | 8 | 1 | 1 |
| OOM Error | | every_layer | torch | 4 | 1 | 1 |
| OOM Error | | disabled | torch | 4 | 1 | 1 |
| OOM Error | | disabled | fused | 4 | 1 | 1 |
| OOM Error | | disabled | flash_attn1.0.8 | 4 | 1 | 1 |
| OOM Error | | disabled | torch | 2 | 1 | 1 |
| OOM Error | | disabled | fused | 2 | 1 | 1 |
| OOM Error | | disabled | flash_attn1.0.8 | 2 | 1 | 1 |
| OOM Error | | disabled | torch | 1 | 1 | 1 |
| OOM Error | | disabled | fused | 1 | 1 | 1 |
| OOM Error | | disabled | flash_attn1.0.8 | 1 | 1 | 1 |

## B.3   LLAMA 13B with 8k sequence length

Table 5: Performance analysis of a LLAMA 13B model with a sequence length of 8k trained on 128 A100 GPUs with the AA-Scaling codebase. All measurements use the FLASHATTENTION or PyTorch kernel. The analysis also includes Out of Memory (OOM) error occurrences.

| Step Time | MFU | Activation | Kernel | MB Size | TP Size | PP Size |
|---|---|---|---|---|---|---|
| 18.41 | 59.41 | disabled | flash_attn2 + RMS kern. | 1 | 2 | 2 |
| 19.32 | 56.61 | disabled | flash_attn2 + RMS kern. | 1 | 2 | 4 |
| 21.36 | 51.21 | disabled | flash_attn2 + RMS kern. | 1 | 4 | 1 |
| 21.94 | 49.93 | disabled | flash_attn2 + RMS kern. | 1 | 4 | 2 |
| 21.93 | 49.88 | disabled | flash_attn2 | 1 | 2 | 4 |
| 23.46 | 46.90 | disabled | flash_attn2 + RMS kern. | 1 | 4 | 4 |
| 23.78 | 46.27 | disabled | flash_attn2 + RMS kern. | 2 | 4 | 4 |
| 24.62 | 44.42 | every_layer | flash_attn2 | 1 | 1 | 2 |
| 24.84 | 44.03 | disabled | flash_attn1.0.8 | 1 | 2 | 4 |
| 25.99 | 42.08 | disabled | flash_attn2 | 1 | 4 | 2 |
| 26.29 | 41.81 | every_layer | flash_attn2 | 1 | 1 | 4 |
| 26.63 | 41.08 | every_layer | flash_attn2 | 1 | 2 | 1 |
| 26.94 | 40.59 | disabled | flash_attn2 | 1 | 4 | 4 |
| 27.42 | 39.89 | every_layer | flash_attn2 | 1 | 2 | 2 |
| 27.53 | 39.73 | every_layer | flash_attn2 | 2 | 1 | 2 |
| 28.08 | 38.94 | every_layer | flash_attn2 | 1 | 2 | 4 |
| 28.15 | 38.85 | every_layer | flash_attn2 | 1 | 1 | 1 |
| 28.52 | 38.34 | disabled | flash_attn1.0.8 | 1 | 4 | 2 |
| 28.73 | 38.06 | every_layer | flash_attn1.0.8 | 1 | 1 | 2 |
| 29.08 | 37.60 | every_layer | flash_attn2 | 2 | 1 | 4 |
| 29.19 | 37.49 | every_layer | flash_attn2 | 2 | 2 | 2 |
| 29.21 | 37.43 | disabled | flash_attn1.0.8 | 1 | 4 | 4 |
| 30.03 | 36.42 | every_layer | flash_attn1.0.8 | 1 | 1 | 4 |
| 30.08 | 36.36 | every_layer | flash_attn2 | 2 | 2 | 4 |
| 30.14 | 36.28 | every_layer | flash_attn1.0.8 | 1 | 2 | 1 |
| 31.65 | 34.55 | every_layer | flash_attn1.0.8 | 2 | 1 | 2 |
| 31.86 | 34.32 | every_layer | flash_attn1.0.8 | 1 | 2 | 2 |
| 32.13 | 34.04 | every_layer | flash_attn2 | 2 | 4 | 1 |
| 32.57 | 33.58 | every_layer | flash_attn2 | 1 | 4 | 1 |
| 32.91 | 33.23 | every_layer | flash_attn1.0.8 | 2 | 2 | 2 |
| 34.75 | 32.47 | every_layer | flash_attn1.0.8 | 1 | 2 | 4 |
| 34.33 | 31.85 | every_layer | flash_attn1.0.8 | 2 | 1 | 4 |
| 34.61 | 31.60 | every_layer | flash_attn1.0.8 | 2 | 2 | 4 |
| 34.64 | 31.58 | every_layer | flash_attn2 | 2 | 4 | 2 |
| 34.76 | 31.46 | every_layer | flash_attn2 | 1 | 4 | 2 |
| 35.86 | 30.49 | every_layer | flash_attn1.0.8 | 2 | 4 | 1 |
| 36.07 | 30.32 | every_layer | flash_attn2 | 2 | 4 | 4 |
| 36.33 | 30.10 | every_layer | flash_attn1.0.8 | 1 | 4 | 1 |
| 36.71 | 29.92 | every_layer | flash_attn2 | 1 | 4 | 4 |
| 37.63 | 29.06 | every_layer | flash_attn1.0.8 | 1 | 4 | 2 |
| 38.26 | 28.59 | every_layer | flash_attn1.0.8 | 2 | 4 | 2 |
| 38.61 | 28.33 | every_layer | flash_attn1.0.8 | 1 | 4 | 4 |
| 39.76 | 27.50 | every_layer | flash_attn1.0.8 | 2 | 4 | 4 |
| 51.95 | 21.05 | every_layer | torch | 1 | 2 | 1 |
| 54.97 | 19.89 | every_layer | torch | 1 | 2 | 2 |
| 55.08 | 19.85 | every_layer | torch | 1 | 1 | 2 |
| 58.37 | 18.73 | every_layer | torch | 1 | 4 | 1 |
| 58.61 | 18.66 | every_layer | torch | 1 | 2 | 4 |
| 59.00 | 18.56 | every_layer | torch | 1 | 1 | 4 |
| 60.64 | 18.03 | every_layer | torch | 1 | 4 | 2 |
| 62.00 | 17.64 | every_layer | torch | 2 | 4 | 2 |
| 64.25 | 17.02 | every_layer | torch | 2 | 2 | 4 |

| Step Time | MFU | Activation | Kernel | MB Size | TP Size | PP Size |
|---|---|---|---|---|---|---|
| 64.48 | 16.96 | every_layer | torch | 1 | 4 | 4 |
| 66.48 | 16.45 | every_layer | torch | 2 | 4 | 4 |
| OOM Error | | disabled | flash_attn2 + RMS kern. | 4 | 4 | 1 |
| OOM Error | | disabled | flash_attn2 + RMS kern. | 2 | 4 | 2 |
| OOM Error | | disabled | flash_attn2 + RMS kern. | 4 | 2 | 4 |
| OOM Error | | disabled | flash_attn2 + RMS kern. | 4 | 4 | 2 |
| OOM Error | | disabled | flash_attn2 + RMS kern. | 2 | 4 | 1 |
| OOM Error | | disabled | flash_attn2 + RMS kern. | 2 | 2 | 4 |
| OOM Error | | disabled | flash_attn2 + RMS kern. | 4 | 1 | 1 |
| OOM Error | | disabled | flash_attn2 + RMS kern. | 4 | 2 | 2 |
| OOM Error | | disabled | flash_attn2 + RMS kern. | 4 | 1 | 2 |
| OOM Error | | disabled | flash_attn2 + RMS kern. | 2 | 2 | 2 |
| OOM Error | | disabled | flash_attn2 + RMS kern. | 2 | 2 | 1 |
| OOM Error | | disabled | flash_attn2 + RMS kern. | 4 | 2 | 1 |
| OOM Error | | disabled | flash_attn2 + RMS kern. | 4 | 1 | 4 |
| OOM Error | | disabled | flash_attn2 + RMS kern. | 2 | 1 | 4 |
| OOM Error | | disabled | flash_attn2 + RMS kern. | 2 | 1 | 2 |
| OOM Error | | disabled | flash_attn2 + RMS kern. | 2 | 1 | 1 |
| OOM Error | | disabled | flash_attn2 + RMS kern. | 1 | 2 | 1 |
| OOM Error | | disabled | flash_attn2 + RMS kern. | 1 | 1 | 4 |
| OOM Error | | disabled | flash_attn2 + RMS kern. | 1 | 1 | 2 |
| OOM Error | | disabled | flash_attn2 + RMS kern. | 1 | 1 | 1 |
| OOM Error | | every_layer | flash_attn2 | 4 | 1 | 1 |
| OOM Error | | every_layer | flash_attn2 | 4 | 4 | 1 |
| OOM Error | | every_layer | flash_attn2 | 4 | 2 | 2 |
| OOM Error | | every_layer | flash_attn2 | 4 | 4 | 2 |
| OOM Error | | every_layer | flash_attn2 | 4 | 1 | 2 |
| OOM Error | | every_layer | flash_attn2 | 4 | 4 | 4 |
| OOM Error | | every_layer | flash_attn2 | 2 | 2 | 1 |
| OOM Error | | every_layer | flash_attn2 | 4 | 2 | 1 |
| OOM Error | | every_layer | flash_attn2 | 4 | 1 | 4 |
| OOM Error | | every_layer | flash_attn2 | 4 | 2 | 4 |
| OOM Error | | every_layer | flash_attn2 | 2 | 1 | 1 |
| OOM Error | | disabled | flash_attn2 | 4 | 4 | 4 |
| OOM Error | | disabled | flash_attn2 | 4 | 4 | 2 |
| OOM Error | | disabled | flash_attn2 | 4 | 4 | 1 |
| OOM Error | | disabled | flash_attn2 | 4 | 2 | 4 |
| OOM Error | | disabled | flash_attn2 | 4 | 2 | 2 |
| OOM Error | | disabled | flash_attn2 | 4 | 2 | 1 |
| OOM Error | | disabled | flash_attn2 | 4 | 1 | 4 |
| OOM Error | | disabled | flash_attn2 | 4 | 1 | 2 |
| OOM Error | | disabled | flash_attn2 | 4 | 1 | 1 |
| OOM Error | | disabled | flash_attn2 | 2 | 4 | 4 |
| OOM Error | | disabled | flash_attn2 | 2 | 4 | 2 |
| OOM Error | | disabled | flash_attn2 | 2 | 4 | 1 |
| OOM Error | | disabled | flash_attn2 | 2 | 2 | 4 |
| OOM Error | | disabled | flash_attn2 | 2 | 2 | 2 |
| OOM Error | | disabled | flash_attn2 | 2 | 2 | 1 |
| OOM Error | | disabled | flash_attn2 | 2 | 1 | 4 |
| OOM Error | | disabled | flash_attn2 | 2 | 1 | 2 |
| OOM Error | | disabled | flash_attn2 | 2 | 1 | 1 |
| OOM Error | | disabled | flash_attn2 | 1 | 4 | 1 |
| OOM Error | | disabled | flash_attn2 | 1 | 2 | 2 |
| OOM Error | | disabled | flash_attn2 | 1 | 2 | 1 |
| OOM Error | | disabled | flash_attn2 | 1 | 1 | 4 |
| OOM Error | | disabled | flash_attn2 | 1 | 1 | 2 |
| OOM Error | | disabled | flash_attn2 | 1 | 1 | 1 |
| OOM Error | | every_layer | flash_attn1.0.8 | 4 | 4 | 4 |
| OOM Error | | every_layer | flash_attn1.0.8 | 4 | 4 | 1 |
| OOM Error | | every_layer | flash_attn1.0.8 | 4 | 2 | 2 |
| OOM Error | | every_layer | flash_attn1.0.8 | 4 | 1 | 4 |
| OOM Error | | every_layer | flash_attn1.0.8 | 2 | 2 | 1 |
| OOM Error | | every_layer | flash_attn1.0.8 | 4 | 1 | 1 |
| OOM Error | | every_layer | flash_attn1.0.8 | 4 | 2 | 1 |
| OOM Error | | every_layer | flash_attn1.0.8 | 4 | 4 | 2 |
| OOM Error | | every_layer | flash_attn1.0.8 | 4 | 1 | 2 |
| OOM Error | | every_layer | flash_attn1.0.8 | 4 | 2 | 4 |
| OOM Error | | every_layer | flash_attn1.0.8 | 2 | 1 | 1 |
| OOM Error | | every_layer | flash_attn1.0.8 | 1 | 1 | 1 |
| OOM Error | | disabled | flash_attn1.0.8 | 4 | 4 | 4 |
| OOM Error | | disabled | flash_attn1.0.8 | 4 | 4 | 2 |
| OOM Error | | disabled | flash_attn1.0.8 | 4 | 4 | 1 |
| OOM Error | | disabled | flash_attn1.0.8 | 4 | 2 | 4 |
| OOM Error | | disabled | flash_attn1.0.8 | 4 | 2 | 2 |
| OOM Error | | disabled | flash_attn1.0.8 | 4 | 2 | 1 |
| OOM Error | | disabled | flash_attn1.0.8 | 4 | 1 | 4 |
| OOM Error | | disabled | flash_attn1.0.8 | 4 | 1 | 2 |
| OOM Error | | disabled | flash_attn1.0.8 | 4 | 1 | 1 |
| OOM Error | | disabled | flash_attn1.0.8 | 2 | 4 | 4 |

| Step Time | MFU | Activation | Kernel | MB Size | TP Size | PP Size |
|---|---|---|---|---|---|---|
| OOM Error | | disabled | flash_attn1.0.8 | 2 | 4 | 2 |
| OOM Error | | disabled | flash_attn1.0.8 | 2 | 4 | 1 |
| OOM Error | | disabled | flash_attn1.0.8 | 2 | 2 | 4 |
| OOM Error | | disabled | flash_attn1.0.8 | 2 | 2 | 2 |
| OOM Error | | disabled | flash_attn1.0.8 | 2 | 2 | 1 |
| OOM Error | | disabled | flash_attn1.0.8 | 2 | 1 | 4 |
| OOM Error | | disabled | flash_attn1.0.8 | 2 | 1 | 2 |
| OOM Error | | disabled | flash_attn1.0.8 | 2 | 1 | 1 |
| OOM Error | | disabled | flash_attn1.0.8 | 1 | 4 | 1 |
| OOM Error | | disabled | flash_attn1.0.8 | 1 | 2 | 2 |
| OOM Error | | disabled | flash_attn1.0.8 | 1 | 2 | 1 |
| OOM Error | | disabled | flash_attn1.0.8 | 1 | 1 | 4 |
| OOM Error | | disabled | flash_attn1.0.8 | 1 | 1 | 2 |
| OOM Error | | disabled | flash_attn1.0.8 | 1 | 1 | 1 |
| OOM Error | | every_layer | torch | 4 | 4 | 4 |
| OOM Error | | every_layer | torch | 4 | 2 | 2 |
| OOM Error | | every_layer | torch | 4 | 4 | 2 |
| OOM Error | | every_layer | torch | 4 | 4 | 1 |
| OOM Error | | every_layer | torch | 4 | 1 | 2 |
| OOM Error | | every_layer | torch | 4 | 2 | 4 |
| OOM Error | | every_layer | torch | 2 | 4 | 1 |
| OOM Error | | every_layer | torch | 2 | 1 | 4 |
| OOM Error | | every_layer | torch | 2 | 2 | 2 |
| OOM Error | | every_layer | torch | 4 | 1 | 1 |
| OOM Error | | every_layer | torch | 4 | 1 | 4 |
| OOM Error | | every_layer | torch | 4 | 2 | 1 |
| OOM Error | | every_layer | torch | 2 | 2 | 1 |
| OOM Error | | every_layer | torch | 2 | 1 | 2 |
| OOM Error | | every_layer | torch | 2 | 1 | 1 |
| OOM Error | | every_layer | torch | 1 | 1 | 1 |
| OOM Error | | disabled | torch | 4 | 4 | 4 |
| OOM Error | | disabled | torch | 4 | 4 | 2 |
| OOM Error | | disabled | torch | 4 | 4 | 1 |
| OOM Error | | disabled | torch | 4 | 2 | 4 |
| OOM Error | | disabled | torch | 4 | 2 | 2 |
| OOM Error | | disabled | torch | 4 | 2 | 1 |
| OOM Error | | disabled | torch | 4 | 1 | 4 |
| OOM Error | | disabled | torch | 4 | 1 | 2 |
| OOM Error | | disabled | torch | 4 | 1 | 1 |
| OOM Error | | disabled | torch | 2 | 4 | 4 |
| OOM Error | | disabled | torch | 2 | 4 | 2 |
| OOM Error | | disabled | torch | 2 | 4 | 1 |
| OOM Error | | disabled | torch | 2 | 2 | 4 |
| OOM Error | | disabled | torch | 2 | 2 | 2 |
| OOM Error | | disabled | torch | 2 | 2 | 1 |
| OOM Error | | disabled | torch | 2 | 1 | 4 |
| OOM Error | | disabled | torch | 2 | 1 | 2 |
| OOM Error | | disabled | torch | 2 | 1 | 1 |
| OOM Error | | disabled | torch | 1 | 4 | 4 |
| OOM Error | | disabled | torch | 1 | 4 | 2 |
| OOM Error | | disabled | torch | 1 | 4 | 1 |
| OOM Error | | disabled | torch | 1 | 2 | 4 |
| OOM Error | | disabled | torch | 1 | 2 | 2 |
| OOM Error | | disabled | torch | 1 | 2 | 1 |
| OOM Error | | disabled | torch | 1 | 1 | 4 |
| OOM Error | | disabled | torch | 1 | 1 | 2 |
| OOM Error | | disabled | torch | 1 | 1 | 1 |

## B.4   LLAMA 30B

Table 6: Performance analysis of a LLAMA 30B model trained on 256 A100 GPUs with the AA-Scaling codebase. The analysis also includes Out of Memory (OOM) error occurrences and errors where the fused kernel does not support this specific tensor parallel configuration.

| Step Time | MFU | Activation | Kernel | MB Size | TP Size | PP Size |
|---|---|---|---|---|---|---|
| 22.67 | 49.22 | disabled | flash_attn2 + RMS kern. | 1 | 2 | 4 |
| 24.00 | 46.76 | disabled | flash_attn2 + RMS kern. | 1 | 1 | 4 |
| 24.26 | 46.01 | disabled | flash_attn2 + RMS kern. | 2 | 2 | 4 |
| 24.70 | 45.16 | disabled | flash_attn2 | 1 | 2 | 4 |
| 25.05 | 44.55 | disabled | flash_attn2 + RMS kern. | 4 | 4 | 4 |
| 25.32 | 44.06 | disabled | flash_attn2 + RMS kern. | 2 | 4 | 4 |
| 26.06 | 42.80 | disabled | flash_attn1.0.8 | 1 | 2 | 4 |
| 26.26 | 42.48 | disabled | flash_attn2 + RMS kern. | 2 | 4 | 2 |
| 26.28 | 42.45 | disabled | flash_attn2 + RMS kern. | 1 | 4 | 4 |

| Step Time | MFU | Activation | Kernel | MB Size | TP Size | PP Size |
|---|---|---|---|---|---|---|
| 26.94 | 41.42 | disabled | flash_attn2 + RMS kern. | 1 | 4 | 2 |
| 28.34 | 39.39 | disabled | flash_attn2 + RMS kern. | 1 | 2 | 2 |
| 29.08 | 38.37 | every_layer | flash_attn2 | 1 | 1 | 4 |
| 29.13 | 38.32 | disabled | flash_attn2 | 1 | 2 | 2 |
| 29.16 | 38.26 | every_layer | flash_attn1.0.8 | 1 | 1 | 4 |
| 29.45 | 37.88 | disabled | flash_attn2 | 2 | 4 | 4 |
| 29.55 | 37.75 | disabled | flash_attn1.0.8 | 2 | 4 | 4 |
| 29.64 | 37.67 | disabled | flash_attn1.0.8 | 1 | 2 | 2 |
| 29.76 | 37.48 | every_layer | flash_attn2 | 2 | 1 | 4 |
| 29.96 | 37.23 | every_layer | flash_attn1.0.8 | 2 | 1 | 4 |
| 30.08 | 37.09 | disabled | flash_attn2 | 1 | 4 | 4 |
| 31.13 | 35.84 | every_layer | flash_attn2 | 2 | 2 | 4 |
| 31.38 | 35.54 | disabled | flash_attn1.0.8 | 1 | 4 | 4 |
| 31.47 | 35.46 | disabled | flash_attn2 | 1 | 4 | 2 |
| 31.66 | 35.24 | disabled | flash_attn2 | 2 | 4 | 2 |
| 31.77 | 35.15 | disabled | flash_attn2 + RMS kern. | 1 | 4 | 1 |
| 31.81 | 35.08 | every_layer | flash_attn2 | 1 | 2 | 4 |
| 31.94 | 34.93 | every_layer | flash_attn2 | 1 | 2 | 2 |
| 32.17 | 34.68 | every_layer | flash_attn1.0.8 | 2 | 2 | 4 |
| 32.27 | 34.57 | every_layer | flash_attn2 | 2 | 2 | 2 |
| 32.42 | 34.41 | disabled | flash_attn1.0.8 | 1 | 4 | 2 |
| 32.73 | 34.08 | every_layer | fused | 1 | 1 | 4 |
| 32.84 | 33.96 | every_layer | flash_attn1.0.8 | 1 | 2 | 2 |
| 33.08 | 33.72 | every_layer | fused | 2 | 1 | 4 |
| 33.29 | 33.51 | every_layer | flash_attn1.0.8 | 1 | 2 | 4 |
| 33.47 | 33.33 | every_layer | flash_attn1.0.8 | 4 | 2 | 2 |
| 33.70 | 33.11 | every_layer | flash_attn2 | 4 | 2 | 4 |
| 33.75 | 33.05 | every_layer | flash_attn1.0.8 | 4 | 1 | 4 |
| 33.79 | 33.01 | every_layer | flash_attn1.0.8 | 4 | 2 | 4 |
| 33.97 | 32.85 | disabled | flash_attn1.0.8 | 2 | 4 | 2 |
| 34.22 | 32.61 | every_layer | flash_attn2 | 4 | 1 | 4 |
| 34.45 | 32.39 | every_layer | flash_attn2 | 4 | 2 | 2 |
| 35.07 | 31.84 | every_layer | flash_attn2 | 1 | 1 | 2 |
| 35.23 | 31.67 | every_layer | flash_attn1.0.8 | 2 | 2 | 2 |
| 35.32 | 31.61 | disabled | flash_attn2 + RMS kern. | 2 | 4 | 1 |
| 35.88 | 31.09 | every_layer | fused | 2 | 2 | 4 |
| 36.42 | 30.66 | every_layer | flash_attn1.0.8 | 1 | 1 | 2 |
| 36.48 | 30.58 | every_layer | fused | 4 | 2 | 4 |
| 37.12 | 30.06 | every_layer | flash_attn2 | 4 | 4 | 2 |
| 37.76 | 29.58 | every_layer | flash_attn2 | 2 | 1 | 2 |
| 37.81 | 29.50 | every_layer | flash_attn1.0.8 | 4 | 4 | 2 |
| 37.94 | 29.40 | every_layer | fused | 4 | 1 | 4 |
| 37.99 | 29.38 | every_layer | fused | 1 | 1 | 2 |
| 38.07 | 29.32 | every_layer | flash_attn1.0.8 | 2 | 1 | 2 |
| 38.28 | 29.15 | every_layer | flash_attn2 | 4 | 4 | 4 |
| 38.51 | 28.97 | every_layer | flash_attn2 | 2 | 4 | 4 |
| 38.60 | 28.90 | every_layer | flash_attn1.0.8 | 4 | 4 | 4 |
| 39.57 | 28.19 | every_layer | flash_attn2 | 2 | 4 | 2 |
| 39.61 | 28.17 | every_layer | fused | 4 | 2 | 2 |
| 39.65 | 28.14 | every_layer | flash_attn1.0.8 | 2 | 4 | 4 |
| 39.70 | 28.10 | every_layer | flash_attn2 | 1 | 2 | 1 |
| 39.74 | 28.07 | every_layer | fused | 2 | 1 | 2 |
| 39.81 | 28.03 | disabled | flash_attn2 | 1 | 4 | 1 |
| 40.07 | 27.84 | every_layer | flash_attn2 | 1 | 4 | 2 |
| 40.22 | 27.73 | every_layer | fused | 4 | 4 | 2 |
| 40.29 | 27.68 | every_layer | flash_attn1.0.8 | 2 | 4 | 2 |
| 41.01 | 27.20 | every_layer | flash_attn2 | 1 | 4 | 4 |
| 41.74 | 26.73 | every_layer | fused | 4 | 4 | 4 |
| 41.80 | 26.69 | every_layer | flash_attn1.0.8 | 1 | 2 | 1 |
| 41.84 | 26.66 | every_layer | flash_attn1.0.8 | 4 | 4 | 1 |
| 41.94 | 26.60 | every_layer | flash_attn1.0.8 | 1 | 4 | 4 |
| 42.13 | 26.48 | every_layer | flash_attn2 | 2 | 4 | 1 |
| 42.22 | 26.42 | every_layer | fused | 4 | 4 | 1 |
| 42.41 | 26.30 | every_layer | flash_attn1.0.8 | 1 | 4 | 2 |
| 43.24 | 25.80 | disabled | flash_attn1.0.8 | 1 | 4 | 1 |
| 43.51 | 25.64 | every_layer | flash_attn1.0.8 | 2 | 4 | 1 |
| 43.92 | 25.40 | every_layer | flash_attn2 | 4 | 4 | 1 |
| 44.09 | 25.30 | every_layer | flash_attn2 | 1 | 4 | 1 |
| 45.56 | 24.49 | every_layer | flash_attn1.0.8 | 1 | 4 | 1 |
| OOM Error | | disabled | flash_attn2 + RMS kern. | 4 | 2 | 4 |
| OOM Error | | disabled | flash_attn2 + RMS kern. | 4 | 4 | 1 |
| OOM Error | | disabled | flash_attn2 + RMS kern. | 2 | 2 | 1 |
| OOM Error | | disabled | flash_attn2 + RMS kern. | 4 | 1 | 2 |
| OOM Error | | disabled | flash_attn2 + RMS kern. | 4 | 1 | 1 |
| OOM Error | | disabled | flash_attn2 + RMS kern. | 4 | 4 | 2 |
| OOM Error | | disabled | flash_attn2 + RMS kern. | 2 | 2 | 2 |
| OOM Error | | disabled | flash_attn2 + RMS kern. | 4 | 2 | 2 |
| OOM Error | | disabled | flash_attn2 + RMS kern. | 2 | 1 | 4 |
| OOM Error | | disabled | flash_attn2 + RMS kern. | 4 | 1 | 4 |

| Step Time | MFU | Activation | Kernel | MB Size | TP Size | PP Size |
|---|---|---|---|---|---|---|
| OOM Error | | disabled | flash_attn2 + RMS kern. | 2 | 1 | 2 |
| OOM Error | | disabled | flash_attn2 + RMS kern. | 2 | 1 | 1 |
| OOM Error | | disabled | flash_attn2 + RMS kern. | 1 | 2 | 1 |
| OOM Error | | disabled | flash_attn2 + RMS kern. | 1 | 1 | 2 |
| OOM Error | | disabled | flash_attn2 + RMS kern. | 1 | 1 | 1 |
| OOM Error | | every_layer | flash_attn2 | 2 | 2 | 1 |
| OOM Error | | every_layer | flash_attn2 | 4 | 2 | 1 |
| OOM Error | | every_layer | flash_attn2 | 4 | 1 | 2 |
| OOM Error | | every_layer | flash_attn2 | 4 | 1 | 1 |
| OOM Error | | every_layer | flash_attn2 | 2 | 1 | 1 |
| OOM Error | | every_layer | flash_attn2 | 1 | 1 | 1 |
| OOM Error | | disabled | flash_attn2 | 4 | 4 | 4 |
| OOM Error | | disabled | flash_attn2 | 4 | 4 | 2 |
| OOM Error | | disabled | flash_attn2 | 4 | 4 | 1 |
| OOM Error | | disabled | flash_attn2 | 4 | 2 | 4 |
| OOM Error | | disabled | flash_attn2 | 4 | 2 | 2 |
| OOM Error | | disabled | flash_attn2 | 4 | 2 | 1 |
| OOM Error | | disabled | flash_attn2 | 4 | 1 | 4 |
| OOM Error | | disabled | flash_attn2 | 4 | 1 | 2 |
| OOM Error | | disabled | flash_attn2 | 4 | 1 | 1 |
| OOM Error | | disabled | flash_attn2 | 2 | 4 | 1 |
| OOM Error | | disabled | flash_attn2 | 2 | 2 | 4 |
| OOM Error | | disabled | flash_attn2 | 2 | 2 | 2 |
| OOM Error | | disabled | flash_attn2 | 2 | 2 | 1 |
| OOM Error | | disabled | flash_attn2 | 2 | 1 | 4 |
| OOM Error | | disabled | flash_attn2 | 2 | 1 | 2 |
| OOM Error | | disabled | flash_attn2 | 2 | 1 | 1 |
| OOM Error | | disabled | flash_attn2 | 1 | 2 | 1 |
| OOM Error | | disabled | flash_attn2 | 1 | 1 | 4 |
| OOM Error | | disabled | flash_attn2 | 1 | 1 | 2 |
| OOM Error | | disabled | flash_attn2 | 1 | 1 | 1 |
| OOM Error | | every_layer | fused | 4 | 2 | 1 |
| OOM Error | | every_layer | fused | 4 | 1 | 2 |
| OOM Error | | every_layer | fused | 4 | 1 | 1 |
| OOM Error | | every_layer | fused | 2 | 2 | 1 |
| OOM Error | | every_layer | fused | 2 | 1 | 1 |
| OOM Error | | every_layer | fused | 1 | 1 | 1 |
| OOM Error | | every_layer | flash_attn1.0.8 | 4 | 2 | 1 |
| OOM Error | | every_layer | flash_attn1.0.8 | 4 | 1 | 2 |
| OOM Error | | every_layer | flash_attn1.0.8 | 4 | 1 | 1 |
| OOM Error | | every_layer | flash_attn1.0.8 | 2 | 2 | 1 |
| OOM Error | | every_layer | flash_attn1.0.8 | 2 | 1 | 1 |
| OOM Error | | every_layer | flash_attn1.0.8 | 1 | 1 | 1 |
| OOM Error | | disabled | fused | 4 | 4 | 4 |
| OOM Error | | disabled | fused | 4 | 4 | 2 |
| OOM Error | | disabled | fused | 4 | 4 | 1 |
| OOM Error | | disabled | fused | 4 | 2 | 4 |
| OOM Error | | disabled | fused | 4 | 2 | 2 |
| OOM Error | | disabled | fused | 4 | 2 | 1 |
| OOM Error | | disabled | fused | 4 | 1 | 4 |
| OOM Error | | disabled | fused | 4 | 1 | 2 |
| OOM Error | | disabled | fused | 4 | 1 | 1 |
| OOM Error | | disabled | fused | 2 | 2 | 4 |
| OOM Error | | disabled | fused | 2 | 2 | 2 |
| OOM Error | | disabled | fused | 2 | 2 | 1 |
| OOM Error | | disabled | fused | 2 | 1 | 4 |
| OOM Error | | disabled | fused | 2 | 1 | 2 |
| OOM Error | | disabled | flash_attn1.0.8 | 4 | 4 | 4 |
| OOM Error | | disabled | flash_attn1.0.8 | 4 | 4 | 2 |
| OOM Error | | disabled | flash_attn1.0.8 | 4 | 4 | 1 |
| OOM Error | | disabled | flash_attn1.0.8 | 4 | 2 | 4 |
| OOM Error | | disabled | flash_attn1.0.8 | 4 | 2 | 2 |
| OOM Error | | disabled | flash_attn1.0.8 | 4 | 2 | 1 |
| OOM Error | | disabled | flash_attn1.0.8 | 4 | 1 | 4 |
| OOM Error | | disabled | flash_attn1.0.8 | 4 | 1 | 2 |
| OOM Error | | disabled | flash_attn1.0.8 | 4 | 1 | 1 |
| OOM Error | | disabled | flash_attn1.0.8 | 2 | 4 | 1 |
| OOM Error | | disabled | flash_attn1.0.8 | 2 | 2 | 4 |
| OOM Error | | disabled | flash_attn1.0.8 | 2 | 2 | 2 |
| OOM Error | | disabled | flash_attn1.0.8 | 2 | 2 | 1 |
| OOM Error | | disabled | flash_attn1.0.8 | 2 | 1 | 4 |
| OOM Error | | disabled | flash_attn1.0.8 | 2 | 1 | 2 |
| OOM Error | | disabled | flash_attn1.0.8 | 2 | 1 | 1 |
| OOM Error | | disabled | flash_attn1.0.8 | 1 | 2 | 1 |
| OOM Error | | disabled | flash_attn1.0.8 | 1 | 1 | 4 |
| OOM Error | | disabled | flash_attn1.0.8 | 1 | 1 | 2 |
| OOM Error | | disabled | flash_attn1.0.8 | 1 | 1 | 1 |
| Kernel unavail. | | every_layer | fused | 2 | 4 | 4 |
| Kernel unavail. | | every_layer | fused | 2 | 4 | 2 |

| Step Time | MFU | Activation | Kernel | MB Size | TP Size | PP Size |
|---|---|---|---|---|---|---|
| Kernel unavail. | | every_layer | fused | 2 | 4 | 1 |
| Kernel unavail. | | every_layer | fused | 2 | 2 | 2 |
| Kernel unavail. | | every_layer | fused | 1 | 4 | 4 |
| Kernel unavail. | | every_layer | fused | 1 | 4 | 2 |
| Kernel unavail. | | every_layer | fused | 1 | 4 | 1 |
| Kernel unavail. | | every_layer | fused | 1 | 2 | 4 |
| Kernel unavail. | | every_layer | fused | 1 | 2 | 2 |
| Kernel unavail. | | every_layer | fused | 1 | 2 | 1 |
| Kernel unavail. | | disabled | fused | 2 | 4 | 4 |
| Kernel unavail. | | disabled | fused | 2 | 4 | 2 |
| Kernel unavail. | | disabled | fused | 2 | 4 | 1 |
| Kernel unavail. | | disabled | fused | 1 | 4 | 4 |
| Kernel unavail. | | disabled | fused | 1 | 4 | 2 |
| Kernel unavail. | | disabled | fused | 1 | 4 | 1 |
| Kernel unavail. | | disabled | fused | 1 | 2 | 4 |
| Kernel unavail. | | disabled | fused | 1 | 2 | 2 |
| Kernel unavail. | | disabled | fused | 1 | 2 | 1 |

## B.5   LLAMA 30B with 8k sequence length

Table 7: Performance analysis of a LLAMA 30B model with 8k sequence length trained on 128 A100 GPUs with the AA-Scaling codebase. All measurements use the FLASHATTENTION kernel. The analysis also includes Out of Memory (OOM) error occurrences.

| Step Time | MFU | Activation | Kernel | MB Size | TP Size | PP Size |
|---|---|---|---|---|---|---|
| 49.43 | 51.40 | disabled | flash_attn2 + RMS kern. | 1 | 4 | 4 |
| 50.23 | 50.57 | disabled | flash_attn2 + RMS kern. | 1 | 4 | 8 |
| 54.78 | 46.37 | disabled | flash_attn2 + RMS kern. | 1 | 4 | 16 |
| 62.84 | 40.43 | every_layer | flash_attn2 | 1 | 2 | 4 |
| 62.99 | 40.33 | every_layer | flash_attn2 | 1 | 2 | 2 |
| 63.99 | 39.70 | every_layer | flash_attn2 | 1 | 2 | 8 |
| 66.69 | 38.09 | every_layer | flash_attn2 | 1 | 2 | 16 |
| 66.71 | 38.08 | every_layer | flash_attn2 | 2 | 2 | 4 |
| 68.99 | 36.82 | every_layer | flash_attn2 | 2 | 2 | 8 |
| 69.45 | 36.58 | every_layer | flash_attn1.0.8 | 1 | 2 | 2 |
| 70.01 | 36.29 | every_layer | flash_attn1.0.8 | 1 | 2 | 4 |
| 71.81 | 35.38 | every_layer | flash_attn2 | 2 | 2 | 16 |
| 71.87 | 35.34 | every_layer | flash_attn1.0.8 | 1 | 2 | 8 |
| 72.42 | 35.08 | every_layer | flash_attn2 | 1 | 4 | 2 |
| 73.33 | 34.64 | every_layer | flash_attn2 | 2 | 4 | 2 |
| 75.02 | 33.86 | every_layer | flash_attn1.0.8 | 2 | 2 | 4 |
| 75.53 | 33.64 | every_layer | flash_attn2 | 1 | 4 | 4 |
| 76.53 | 33.20 | every_layer | flash_attn2 | 2 | 4 | 4 |
| 77.42 | 32.81 | every_layer | flash_attn1.0.8 | 2 | 2 | 8 |
| 77.86 | 32.63 | every_layer | flash_attn2 | 1 | 4 | 8 |
| 79.71 | 31.87 | every_layer | flash_attn1.0.8 | 1 | 4 | 2 |
| 79.74 | 31.86 | every_layer | flash_attn2 | 2 | 4 | 8 |
| 80.25 | 31.65 | every_layer | flash_attn1.0.8 | 2 | 4 | 2 |
| 82.16 | 30.92 | every_layer | flash_attn1.0.8 | 1 | 4 | 4 |
| 82.62 | 30.75 | every_layer | flash_attn2 | 1 | 4 | 16 |
| 84.01 | 30.24 | every_layer | flash_attn1.0.8 | 2 | 4 | 4 |
| 85.34 | 29.77 | every_layer | flash_attn2 | 2 | 4 | 16 |
| 85.53 | 29.70 | every_layer | flash_attn1.0.8 | 1 | 4 | 8 |
| 87.18 | 29.14 | every_layer | flash_attn1.0.8 | 2 | 4 | 8 |
| OOM Error | | disabled | flash_attn2 + RMS kern. | 4 | 4 | 16 |
| OOM Error | | disabled | flash_attn2 + RMS kern. | 2 | 2 | 8 |
| OOM Error | | disabled | flash_attn2 + RMS kern. | 2 | 4 | 8 |
| OOM Error | | disabled | flash_attn2 + RMS kern. | 2 | 2 | 16 |
| OOM Error | | disabled | flash_attn2 + RMS kern. | 1 | 2 | 16 |
| OOM Error | | disabled | flash_attn2 + RMS kern. | 4 | 2 | 16 |
| OOM Error | | disabled | flash_attn2 + RMS kern. | 2 | 4 | 16 |
| OOM Error | | disabled | flash_attn2 + RMS kern. | 4 | 4 | 8 |
| OOM Error | | disabled | flash_attn2 + RMS kern. | 1 | 2 | 8 |
| OOM Error | | every_layer | flash_attn2 | 4 | 4 | 16 |
| OOM Error | | every_layer | flash_attn2 | 4 | 2 | 2 |
| OOM Error | | every_layer | flash_attn2 | 4 | 2 | 4 |
| OOM Error | | every_layer | flash_attn2 | 4 | 2 | 16 |
| OOM Error | | every_layer | flash_attn2 | 4 | 4 | 2 |
| OOM Error | | every_layer | flash_attn2 | 4 | 2 | 8 |
| OOM Error | | every_layer | flash_attn2 | 4 | 4 | 4 |
| OOM Error | | every_layer | flash_attn2 | 2 | 2 | 2 |
| OOM Error | | every_layer | flash_attn2 | 4 | 4 | 8 |
| OOM Error | | disabled | flash_attn2 | 4 | 4 | 16 |
| OOM Error | | disabled | flash_attn2 | 4 | 4 | 8 |

| Step Time | MFU | Activation | Kernel | MB Size | TP Size | PP Size |
|---|---|---|---|---|---|---|
| OOM Error | | disabled | flash_attn2 | 4 | 4 | 4 |
| OOM Error | | disabled | flash_attn2 | 4 | 4 | 2 |
| OOM Error | | disabled | flash_attn2 | 4 | 2 | 16 |
| OOM Error | | disabled | flash_attn2 | 4 | 2 | 8 |
| OOM Error | | disabled | flash_attn2 | 4 | 2 | 4 |
| OOM Error | | disabled | flash_attn2 | 4 | 2 | 2 |
| OOM Error | | disabled | flash_attn2 | 2 | 4 | 16 |
| OOM Error | | disabled | flash_attn2 | 2 | 4 | 8 |
| OOM Error | | disabled | flash_attn2 | 2 | 4 | 4 |
| OOM Error | | disabled | flash_attn2 | 2 | 4 | 2 |
| OOM Error | | disabled | flash_attn2 | 2 | 2 | 16 |
| OOM Error | | disabled | flash_attn2 | 2 | 2 | 8 |
| OOM Error | | disabled | flash_attn2 | 2 | 2 | 4 |
| OOM Error | | disabled | flash_attn2 | 2 | 2 | 2 |
| OOM Error | | disabled | flash_attn2 | 1 | 4 | 16 |
| OOM Error | | disabled | flash_attn2 | 1 | 4 | 8 |
| OOM Error | | disabled | flash_attn2 | 1 | 4 | 4 |
| OOM Error | | disabled | flash_attn2 | 1 | 4 | 2 |
| OOM Error | | disabled | flash_attn2 | 1 | 2 | 16 |
| OOM Error | | disabled | flash_attn2 | 1 | 2 | 8 |
| OOM Error | | disabled | flash_attn2 | 1 | 2 | 4 |
| OOM Error | | disabled | flash_attn2 | 1 | 2 | 2 |
| OOM Error | | disabled | flash_attn1.0.8 | 4 | 4 | 4 |
| OOM Error | | disabled | flash_attn1.0.8 | 4 | 2 | 8 |
| OOM Error | | every_layer | flash_attn1.0.8 | 4 | 2 | 4 |
| OOM Error | | every_layer | flash_attn1.0.8 | 4 | 2 | 2 |
| OOM Error | | every_layer | flash_attn1.0.8 | 4 | 4 | 2 |
| OOM Error | | every_layer | flash_attn1.0.8 | 4 | 4 | 4 |
| OOM Error | | every_layer | flash_attn1.0.8 | 4 | 4 | 8 |
| OOM Error | | every_layer | flash_attn1.0.8 | 4 | 2 | 8 |
| OOM Error | | every_layer | flash_attn1.0.8 | 2 | 2 | 2 |
| OOM Error | | disabled | flash_attn1.0.8 | 4 | 4 | 8 |
| OOM Error | | disabled | flash_attn1.0.8 | 4 | 4 | 2 |
| OOM Error | | disabled | flash_attn1.0.8 | 4 | 2 | 4 |
| OOM Error | | disabled | flash_attn1.0.8 | 4 | 2 | 2 |
| OOM Error | | disabled | flash_attn1.0.8 | 2 | 4 | 8 |
| OOM Error | | disabled | flash_attn1.0.8 | 2 | 4 | 4 |
| OOM Error | | disabled | flash_attn1.0.8 | 2 | 4 | 2 |
| OOM Error | | disabled | flash_attn1.0.8 | 2 | 2 | 8 |
| OOM Error | | disabled | flash_attn1.0.8 | 2 | 2 | 4 |
| OOM Error | | disabled | flash_attn1.0.8 | 2 | 2 | 2 |
| OOM Error | | disabled | flash_attn1.0.8 | 1 | 4 | 8 |
| OOM Error | | disabled | flash_attn1.0.8 | 1 | 4 | 4 |
| OOM Error | | disabled | flash_attn1.0.8 | 1 | 4 | 2 |
| OOM Error | | disabled | flash_attn1.0.8 | 1 | 2 | 8 |
| OOM Error | | disabled | flash_attn1.0.8 | 1 | 2 | 4 |
| OOM Error | | disabled | flash_attn1.0.8 | 1 | 2 | 2 |
| OOM Error | | disabled | flash_attn2 + RMS kern. | 4 | 4 | 4 |
| OOM Error | | disabled | flash_attn2 + RMS kern. | 4 | 2 | 2 |
| OOM Error | | disabled | flash_attn2 + RMS kern. | 2 | 4 | 4 |
| OOM Error | | disabled | flash_attn2 + RMS kern. | 4 | 4 | 2 |
| OOM Error | | disabled | flash_attn2 + RMS kern. | 4 | 2 | 4 |
| OOM Error | | disabled | flash_attn2 + RMS kern. | 2 | 2 | 4 |
| OOM Error | | disabled | flash_attn2 + RMS kern. | 2 | 4 | 2 |
| OOM Error | | disabled | flash_attn2 + RMS kern. | 1 | 2 | 2 |
| OOM Error | | disabled | flash_attn2 + RMS kern. | 1 | 2 | 4 |
| OOM Error | | disabled | flash_attn2 + RMS kern. | 1 | 4 | 2 |
| OOM Error | | disabled | flash_attn2 + RMS kern. | 2 | 2 | 2 |

## B.6  LLAMA 65B

Table 8: Performance analysis of a LLAMA 65B model trained on 128 A100 GPUs with the AA-Scaling codebase. All measurements use the FLASHATTENTION kernel. The analysis also includes Out of Memory (OOM) error occurrences.

| Step Time | MFU | Activation | Kernel | MB Size | TP size | PP Size |
|---|---|---|---|---|---|---|
| 79.31 | 55.26 | disabled | flash_attn2 + RMS kern. | 1 | 2 | 4 |
| 79.54 | 55.10 | disabled | flash_attn2 + RMS kern. | 1 | 2 | 8 |
| 82.88 | 52.88 | disabled | flash_attn2 + RMS kern. | 2 | 4 | 4 |
| 86.55 | 50.63 | disabled | flash_attn2 + RMS kern. | 2 | 4 | 8 |
| 86.61 | 50.60 | disabled | flash_attn2 + RMS kern. | 1 | 4 | 4 |
| 87.14 | 50.30 | disabled | flash_attn2 + RMS kern. | 1 | 4 | 2 |
| 88.16 | 49.71 | disabled | flash_attn2 | 1 | 2 | 8 |
| 92.61 | 47.32 | disabled | flash_attn2 + RMS kern. | 1 | 4 | 8 |

| Step Time | MFU | Activation | Kernel | MB Size | TP size | PP Size |
|---|---|---|---|---|---|---|
| 101.28 | 43.28 | disabled | flash_attn2 + RMS kern. | 2 | 8 | 2 |
| 101.97 | 42.98 | disabled | flash_attn2 | 1 | 4 | 4 |
| 103.17 | 42.48 | disabled | flash_attn2 + RMS kern. | 2 | 8 | 4 |
| 106.59 | 41.11 | disabled | flash_attn1.0.8 | 1 | 4 | 4 |
| 107.39 | 40.81 | every_layer | flash_attn2 | 2 | 2 | 4 |
| 107.85 | 40.64 | disabled | flash_attn2 + RMS kern. | 1 | 8 | 2 |
| 107.97 | 40.59 | disabled | flash_attn2 | 1 | 4 | 8 |
| 108.59 | 40.36 | disabled | flash_attn1.0.8 | 1 | 4 | 8 |
| 109.04 | 40.19 | every_layer | flash_attn2 | 4 | 2 | 4 |
| 110.87 | 39.53 | every_layer | flash_attn1.0.8 | 2 | 2 | 4 |
| 111.35 | 39.36 | every_layer | flash_attn2 | 1 | 2 | 4 |
| 111.83 | 39.19 | disabled | flash_attn2 + RMS kern. | 1 | 8 | 4 |
| 111.83 | 39.19 | every_layer | flash_attn1.0.8 | 4 | 2 | 4 |
| 112.77 | 38.86 | every_layer | flash_attn2 | 2 | 2 | 8 |
| 114.42 | 38.30 | every_layer | flash_attn2 | 4 | 2 | 8 |
| 115.05 | 38.09 | every_layer | flash_attn1.0.8 | 1 | 2 | 4 |
| 115.58 | 37.92 | disabled | flash_attn2 + RMS kern. | 2 | 8 | 8 |
| 116.25 | 37.70 | every_layer | flash_attn1.0.8 | 4 | 2 | 8 |
| 116.27 | 37.69 | every_layer | flash_attn1.0.8 | 2 | 2 | 8 |
| 118.39 | 37.02 | every_layer | flash_attn2 | 1 | 2 | 8 |
| 118.68 | 36.93 | every_layer | flash_attn2 | 1 | 2 | 2 |
| 121.70 | 36.01 | every_layer | flash_attn1.0.8 | 1 | 2 | 8 |
| 121.93 | 35.95 | disabled | flash_attn2 + RMS kern. | 1 | 8 | 8 |
| 125.11 | 35.03 | every_layer | flash_attn2 | 4 | 4 | 4 |
| 126.61 | 34.61 | every_layer | flash_attn2 | 4 | 4 | 2 |
| 128.06 | 34.22 | every_layer | flash_attn1.0.8 | 4 | 4 | 2 |
| 129.68 | 33.80 | every_layer | flash_attn2 | 2 | 4 | 4 |
| 129.92 | 33.73 | every_layer | flash_attn1.0.8 | 4 | 4 | 4 |
| 132.34 | 33.11 | every_layer | flash_attn1.0.8 | 2 | 4 | 2 |
| 133.07 | 32.93 | every_layer | flash_attn1.0.8 | 2 | 4 | 4 |
| 133.56 | 32.81 | every_layer | flash_attn2 | 4 | 4 | 8 |
| 136.43 | 32.12 | every_layer | flash_attn2 | 1 | 4 | 2 |
| 137.73 | 31.82 | every_layer | flash_attn1.0.8 | 4 | 4 | 8 |
| 139.67 | 31.38 | disabled | flash_attn2 | 1 | 8 | 2 |
| 140.15 | 31.28 | disabled | flash_attn1.0.8 | 1 | 8 | 2 |
| 140.12 | 31.28 | every_layer | flash_attn1.0.8 | 1 | 4 | 2 |
| 140.95 | 31.09 | every_layer | flash_attn2 | 1 | 4 | 4 |
| 141.87 | 30.89 | every_layer | flash_attn2 | 2 | 4 | 8 |
| 142.77 | 30.70 | disabled | flash_attn2 | 1 | 8 | 4 |
| 144.59 | 30.31 | every_layer | flash_attn1.0.8 | 1 | 4 | 4 |
| 144.83 | 30.26 | every_layer | flash_attn1.0.8 | 2 | 4 | 8 |
| 151.95 | 28.84 | disabled | flash_attn2 | 1 | 8 | 8 |
| 153.61 | 28.53 | disabled | flash_attn1.0.8 | 1 | 8 | 8 |
| 156.54 | 28.00 | every_layer | flash_attn2 | 1 | 4 | 8 |
| 159.57 | 27.46 | every_layer | flash_attn1.0.8 | 1 | 4 | 8 |
| 164.21 | 26.69 | every_layer | flash_attn2 | 4 | 8 | 2 |
| 168.64 | 25.99 | every_layer | flash_attn1.0.8 | 4 | 8 | 2 |
| 168.97 | 25.94 | every_layer | flash_attn2 | 4 | 8 | 4 |
| 173.16 | 25.31 | every_layer | flash_attn1.0.8 | 4 | 8 | 4 |
| 174.43 | 25.13 | every_layer | flash_attn2 | 2 | 8 | 2 |
| 178.66 | 24.53 | every_layer | flash_attn1.0.8 | 2 | 8 | 2 |
| 180.93 | 24.22 | every_layer | flash_attn2 | 2 | 8 | 4 |
| 184.98 | 23.69 | every_layer | flash_attn2 | 4 | 8 | 8 |
| 187.81 | 23.33 | every_layer | flash_attn1.0.8 | 2 | 8 | 4 |
| 192.53 | 22.76 | every_layer | flash_attn1.0.8 | 4 | 8 | 8 |
| 192.69 | 22.75 | every_layer | flash_attn2 | 1 | 8 | 2 |
| 200.12 | 21.90 | every_layer | flash_attn1.0.8 | 1 | 8 | 2 |
| 203.08 | 21.58 | every_layer | flash_attn2 | 2 | 8 | 8 |
| 209.24 | 20.95 | every_layer | flash_attn2 | 1 | 8 | 4 |
| 210.89 | 20.78 | every_layer | flash_attn1.0.8 | 2 | 8 | 8 |
| 215.12 | 20.37 | every_layer | flash_attn1.0.8 | 1 | 8 | 4 |
| 237.99 | 18.42 | every_layer | flash_attn2 | 1 | 8 | 8 |
| 247.97 | 17.67 | every_layer | flash_attn1.0.8 | 1 | 8 | 8 |
| OOM Error | | disabled | flash_attn2 + RMS kern. | 4 | 8 | 8 |
| OOM Error | | disabled | flash_attn2 + RMS kern. | 4 | 2 | 8 |
| OOM Error | | disabled | flash_attn2 + RMS kern. | 4 | 8 | 4 |
| OOM Error | | disabled | flash_attn2 + RMS kern. | 4 | 4 | 2 |
| OOM Error | | disabled | flash_attn2 + RMS kern. | 4 | 2 | 4 |
| OOM Error | | disabled | flash_attn2 + RMS kern. | 4 | 8 | 2 |
| OOM Error | | disabled | flash_attn2 + RMS kern. | 4 | 2 | 2 |
| OOM Error | | disabled | flash_attn2 + RMS kern. | 4 | 4 | 4 |
| OOM Error | | disabled | flash_attn2 + RMS kern. | 2 | 4 | 2 |
| OOM Error | | disabled | flash_attn2 + RMS kern. | 2 | 2 | 8 |
| OOM Error | | disabled | flash_attn2 + RMS kern. | 2 | 2 | 4 |
| OOM Error | | disabled | flash_attn2 + RMS kern. | 2 | 2 | 2 |
| OOM Error | | disabled | flash_attn2 + RMS kern. | 1 | 2 | 2 |
| OOM Error | | every_layer | flash_attn2 | 4 | 2 | 2 |
| OOM Error | | every_layer | flash_attn2 | 2 | 2 | 2 |
| OOM Error | | disabled | flash_attn2 | 4 | 8 | 8 |

| Step Time | MFU | Activation | Kernel | MB Size | TP size | PP Size |
|---|---|---|---|---|---|---|
| OOM Error | | disabled | flash_attn2 | 4 | 8 | 4 |
| OOM Error | | disabled | flash_attn2 | 4 | 8 | 2 |
| OOM Error | | disabled | flash_attn2 | 4 | 4 | 8 |
| OOM Error | | disabled | flash_attn2 | 4 | 4 | 4 |
| OOM Error | | disabled | flash_attn2 | 4 | 4 | 2 |
| OOM Error | | disabled | flash_attn2 | 4 | 2 | 8 |
| OOM Error | | disabled | flash_attn2 | 4 | 2 | 4 |
| OOM Error | | disabled | flash_attn2 | 4 | 2 | 2 |
| OOM Error | | disabled | flash_attn2 | 2 | 8 | 8 |
| OOM Error | | disabled | flash_attn2 | 2 | 8 | 4 |
| OOM Error | | disabled | flash_attn2 | 2 | 8 | 2 |
| OOM Error | | disabled | flash_attn2 | 2 | 4 | 8 |
| OOM Error | | disabled | flash_attn2 | 2 | 4 | 4 |
| OOM Error | | disabled | flash_attn2 | 2 | 4 | 2 |
| OOM Error | | disabled | flash_attn2 | 2 | 2 | 8 |
| OOM Error | | disabled | flash_attn2 | 2 | 2 | 4 |
| OOM Error | | disabled | flash_attn2 | 2 | 2 | 2 |
| OOM Error | | disabled | flash_attn2 | 1 | 4 | 2 |
| OOM Error | | disabled | flash_attn2 | 1 | 2 | 4 |
| OOM Error | | disabled | flash_attn2 | 1 | 2 | 2 |
| OOM Error | | disabled | flash_attn1.0.8 | 1 | 2 | 8 |
| OOM Error | | disabled | flash_attn1.0.8 | 1 | 2 | 4 |
| OOM Error | | disabled | flash_attn1.0.8 | 1 | 2 | 2 |
| OOM Error | | every_layer | flash_attn1.0.8 | 4 | 2 | 2 |
| OOM Error | | every_layer | flash_attn1.0.8 | 2 | 2 | 2 |
| OOM Error | | disabled | flash_attn1.0.8 | 4 | 8 | 8 |
| OOM Error | | disabled | flash_attn1.0.8 | 4 | 4 | 8 |
| OOM Error | | disabled | flash_attn1.0.8 | 4 | 2 | 4 |
| OOM Error | | disabled | flash_attn1.0.8 | 4 | 4 | 4 |
| OOM Error | | disabled | flash_attn1.0.8 | 2 | 8 | 4 |
| OOM Error | | disabled | flash_attn1.0.8 | 2 | 8 | 8 |
| OOM Error | | disabled | flash_attn1.0.8 | 4 | 2 | 2 |
| OOM Error | | disabled | flash_attn1.0.8 | 2 | 4 | 4 |
| OOM Error | | disabled | flash_attn1.0.8 | 2 | 8 | 2 |
| OOM Error | | disabled | flash_attn1.0.8 | 2 | 4 | 2 |
| OOM Error | | disabled | flash_attn1.0.8 | 4 | 8 | 4 |
| OOM Error | | disabled | flash_attn1.0.8 | 2 | 2 | 8 |
| OOM Error | | disabled | flash_attn1.0.8 | 4 | 4 | 2 |
| OOM Error | | disabled | flash_attn1.0.8 | 4 | 8 | 2 |
| OOM Error | | disabled | flash_attn1.0.8 | 4 | 2 | 8 |
| OOM Error | | disabled | flash_attn1.0.8 | 2 | 4 | 8 |
| OOM Error | | disabled | flash_attn1.0.8 | 2 | 2 | 4 |
| OOM Error | | disabled | flash_attn1.0.8 | 2 | 2 | 2 |
| OOM Error | | disabled | flash_attn1.0.8 | 1 | 4 | 2 |

# C   Sequence Parallelism Training Efficiency Sweep

## C.1   Sweep Configurations

| Model | Seq. Len. | GPUs | TP sizes | PP sizes | MB Sizes | Seq. Parallelism |
|-------|-----------|------|----------|----------|----------|------------------|
| 13B | 2k | 32 | {1, 2} | {1, 2} | {1, 2, 4, 8} | {yes, no} |
| 13B | 8k | 64 | {1, 2, 4} | {1, 2, 4} | {1, 2, 4} | {yes, no} |
| 30B | 2k | 64 | {1, 2, 4} | {1, 2, 4} | {1, 2, 4} | {yes, no} |
| 30B | 8k | 64 | {2, 4} | {2, 4, 8, 16} | {1, 2, 4} | {yes, no} |
| 65B | 2k | 64 | {2, 4, 8} | {2, 4, 8} | {1, 2, 4} | {yes, no} |

Table 9: Search space of additional sequence parallel training efficiency sweep. We sweep over the Cartesian product of all mentioned options similar to Table 1. All runs use FLASHATTENTION-2, the RMSNorm kernel, and do not use activation checkpointing.

## C.2   LLAMA 13B

Table 10: Performance analysis of a LLAMA 13B model trained on 32 A100 GPUs, with and without sequence parallelism. All measurements use FLASHATTENTION-2, the RMS norm kernel, and do not make use of activation checkpointing. The analysis also includes Out of Memory (OOM) error occurrences.

| Step Time | MFU | MB Size | TP size | PP Size | Sequence Parallel |
|-----------|-----|---------|---------|---------|-------------------|
| 53.81 | 69.66 | 1 | 1 | 1 | False |
| 53.99 | 69.45 | 1 | 1 | 1 | True |
| 57.61 | 65.04 | 2 | 2 | 1 | False |
| 58.01 | 64.54 | 1 | 1 | 2 | True |
| 58.62 | 63.88 | 1 | 1 | 2 | False |
| 59.55 | 62.86 | 2 | 1 | 2 | True |
| 61.12 | 62.28 | 2 | 2 | 1 | True |
| 60.23 | 62.15 | 2 | 1 | 2 | False |
| 60.96 | 61.43 | 1 | 2 | 1 | True |
| 62.09 | 60.34 | 1 | 2 | 1 | False |
| 65.04 | 57.55 | 2 | 2 | 2 | True |
| 66.13 | 56.59 | 2 | 2 | 2 | False |
| 66.26 | 56.47 | 1 | 2 | 2 | True |
| 67.47 | 55.44 | 4 | 2 | 2 | True |
| 67.94 | 55.08 | 1 | 2 | 2 | False |
| 68.48 | 54.62 | 4 | 2 | 2 | False |
| OOM Error | | 4 | 1 | 2 | True |
| OOM Error | | 8 | 2 | 2 | True |
| OOM Error | | 4 | 1 | 1 | True |
| OOM Error | | 8 | 1 | 2 | True |
| OOM Error | | 8 | 2 | 1 | True |
| OOM Error | | 4 | 2 | 1 | True |
| OOM Error | | 2 | 1 | 1 | True |
| OOM Error | | 2 | 1 | 1 | False |
| OOM Error | | 8 | 2 | 2 | False |
| OOM Error | | 8 | 2 | 1 | False |
| OOM Error | | 8 | 1 | 1 | False |
| OOM Error | | 4 | 2 | 1 | False |
| OOM Error | | 4 | 1 | 2 | False |
| OOM Error | | 8 | 1 | 2 | False |
| OOM Error | | 4 | 1 | 1 | False |

## C.3   LLAMA 13B 8k sequence length

Table 11: Performance analysis of a LLAMA 13B model with 8k sequence length trained on 64 A100 GPUs, with and without sequence parallelism. All measurements use FLASHATTENTION-2, the RMS norm kernel, and do not make use of activation checkpointing. The analysis also includes Out of Memory (OOM) error occurrences.

| Step Time | MFU | MB Size | TP size | PP Size | Sequence Parallel |
|-----------|-----|---------|---------|---------|-------------------|
| 34.84 | 62.78 | 1 | 2 | 2 | True |

| Step Time | MFU | MB Size | TP size | PP Size | Sequence Parallel |
|---|---|---|---|---|---|
| 34.85 | 62.76 | 1 | 2 | 2 | False |
| 35.80 | 61.10 | 1 | 2 | 4 | True |
| 36.60 | 59.76 | 1 | 2 | 4 | False |
| 36.99 | 59.13 | 1 | 4 | 1 | True |
| 38.85 | 56.31 | 1 | 4 | 2 | True |
| 38.90 | 56.23 | 1 | 4 | 1 | False |
| 40.70 | 53.74 | 1 | 4 | 2 | False |
| 40.82 | 53.58 | 1 | 4 | 4 | True |
| 41.06 | 53.27 | 2 | 4 | 4 | True |
| 43.49 | 50.29 | 1 | 4 | 4 | False |
| 44.03 | 49.68 | 2 | 4 | 4 | False |
| 46.37 | 47.18 | 1 | 8 | 1 | True |
| 46.89 | 46.65 | 2 | 8 | 2 | True |
| 49.96 | 43.79 | 1 | 8 | 2 | True |
| 51.03 | 42.87 | 1 | 8 | 1 | False |
| 51.91 | 42.14 | 2 | 8 | 4 | True |
| 54.78 | 39.93 | 1 | 8 | 4 | True |
| 56.23 | 38.90 | 1 | 8 | 2 | False |
| 59.62 | 36.69 | 2 | 8 | 4 | False |
| 62.04 | 35.26 | 1 | 8 | 4 | False |
| OOM Error | | 4 | 1 | 1 | True |
| OOM Error | | 4 | 8 | 2 | True |
| OOM Error | | 4 | 8 | 4 | True |
| OOM Error | | 4 | 4 | 1 | True |
| OOM Error | | 4 | 8 | 1 | True |
| OOM Error | | 4 | 4 | 4 | True |
| OOM Error | | 4 | 4 | 2 | True |
| OOM Error | | 4 | 2 | 4 | True |
| OOM Error | | 4 | 1 | 2 | True |
| OOM Error | | 4 | 2 | 1 | True |
| OOM Error | | 4 | 1 | 4 | True |
| OOM Error | | 4 | 8 | 2 | False |
| OOM Error | | 4 | 4 | 4 | False |
| OOM Error | | 4 | 4 | 1 | False |
| OOM Error | | 4 | 2 | 4 | False |
| OOM Error | | 4 | 2 | 2 | False |
| OOM Error | | 4 | 1 | 2 | False |
| OOM Error | | 4 | 2 | 1 | False |
| OOM Error | | 4 | 8 | 4 | False |
| OOM Error | | 4 | 8 | 1 | False |
| OOM Error | | 4 | 4 | 2 | False |
| OOM Error | | 4 | 1 | 4 | False |
| OOM Error | | 4 | 1 | 1 | False |
| OOM Error | | 2 | 8 | 1 | True |
| OOM Error | | 2 | 2 | 1 | True |
| OOM Error | | 2 | 2 | 4 | True |
| OOM Error | | 2 | 4 | 2 | True |
| OOM Error | | 2 | 2 | 2 | True |
| OOM Error | | 2 | 4 | 1 | True |
| OOM Error | | 1 | 2 | 1 | True |
| OOM Error | | 2 | 2 | 1 | False |
| OOM Error | | 2 | 2 | 4 | False |
| OOM Error | | 2 | 8 | 2 | False |
| OOM Error | | 2 | 8 | 1 | False |
| OOM Error | | 2 | 4 | 1 | False |
| OOM Error | | 2 | 2 | 2 | False |
| OOM Error | | 2 | 4 | 2 | False |
| OOM Error | | 1 | 2 | 1 | False |

## C.4    LLAMA 30B

Table 12: Performance analysis of a LLAMA 30B model trained on 64 A100 GPUs, with and without sequence parallelism. All measurements use FLASHATTENTION-2, the RMS norm kernel, and do not make use of activation checkpointing. The analysis also includes Out of Memory (OOM) error occurrences.

| Step Time | MFU | MB Size | TP size | PP Size | Sequence Parallel |
|---|---|---|---|---|---|
| 72.00 | 61.98 | 1 | 1 | 4 | False |
| 72.60 | 61.47 | 1 | 1 | 4 | True |
| 72.66 | 61.42 | 2 | 2 | 4 | True |
| 74.86 | 59.61 | 2 | 2 | 4 | False |
| 76.58 | 58.27 | 1 | 2 | 4 | True |
| 76.84 | 58.07 | 1 | 2 | 2 | True |

| Step Time | MFU | MB Size | TP size | PP Size | Sequence Parallel |
|---|---|---|---|---|---|
| 77.59 | 57.52 | 4 | 4 | 4 | True |
| 78.43 | 56.89 | 1 | 2 | 4 | False |
| 80.17 | 55.66 | 1 | 2 | 2 | False |
| 81.06 | 55.05 | 2 | 4 | 2 | True |
| 82.22 | 54.28 | 2 | 4 | 4 | True |
| 83.11 | 53.71 | 4 | 4 | 2 | True |
| 83.76 | 53.30 | 4 | 4 | 2 | True |
| 84.88 | 52.57 | 2 | 4 | 2 | False |
| 85.21 | 52.37 | 2 | 4 | 1 | True |
| 85.47 | 52.21 | 1 | 4 | 2 | True |
| 86.44 | 51.63 | 4 | 4 | 4 | False |
| 86.67 | 51.49 | 4 | 4 | 4 | False |
| 87.68 | 50.89 | 1 | 4 | 4 | True |
| 89.33 | 49.95 | 1 | 4 | 1 | True |
| 89.76 | 49.71 | 2 | 4 | 4 | False |
| 89.86 | 49.66 | 1 | 4 | 2 | False |
| 94.12 | 47.42 | 1 | 4 | 1 | False |
| 95.52 | 46.72 | 1 | 4 | 4 | False |
| OOM Error | | 4 | 4 | 1 | True |
| OOM Error | | 4 | 1 | 1 | True |
| OOM Error | | 4 | 1 | 4 | True |
| OOM Error | | 4 | 2 | 2 | True |
| OOM Error | | 4 | 1 | 2 | True |
| OOM Error | | 4 | 2 | 1 | True |
| OOM Error | | 4 | 2 | 4 | True |
| OOM Error | | 4 | 4 | 1 | True |
| OOM Error | | 2 | 4 | 1 | False |
| OOM Error | | 4 | 4 | 2 | False |
| OOM Error | | 4 | 4 | 1 | False |
| OOM Error | | 4 | 4 | 2 | False |
| OOM Error | | 4 | 2 | 2 | False |
| OOM Error | | 4 | 4 | 1 | False |
| OOM Error | | 4 | 1 | 4 | False |
| OOM Error | | 4 | 2 | 1 | False |
| OOM Error | | 4 | 1 | 2 | False |
| OOM Error | | 4 | 1 | 1 | False |
| OOM Error | | 4 | 2 | 4 | False |
| OOM Error | | 2 | 1 | 2 | True |
| OOM Error | | 2 | 1 | 4 | True |
| OOM Error | | 2 | 2 | 2 | True |
| OOM Error | | 1 | 1 | 2 | True |
| OOM Error | | 2 | 1 | 2 | False |
| OOM Error | | 2 | 1 | 4 | False |
| OOM Error | | 2 | 2 | 2 | False |
| OOM Error | | 1 | 1 | 2 | False |

## C.5   LLAMA 30B 8k sequence length

Table 13: Performance analysis of a LLAMA 30B model with 8k sequence length trained on 64 A100 GPUs, with and without sequence parallelism. All measurements use FLASHATTENTION-2, the RMS norm kernel, and do not make use of activation checkpointing. The analysis also includes Out of Memory (OOM) error occurrences.

| Step Time | MFU | MB Size | TP size | PP Size | Sequence Parallel |
|---|---|---|---|---|---|
| 84.37 | 60.22 | 1 | 4 | 2 | True |
| 85.69 | 59.35 | 1 | 4 | 4 | True |
| 86.16 | 58.97 | 1 | 4 | 16 | True |
| 86.17 | 58.97 | 1 | 4 | 8 | True |
| 93.84 | 54.15 | 1 | 4 | 4 | False |
| 96.70 | 52.54 | 1 | 4 | 8 | False |
| 98.37 | 51.65 | 1 | 4 | 16 | False |
| OOM Error | | 1 | 2 | 2 | True |
| OOM Error | | 2 | 2 | 4 | True |
| OOM Error | | 1 | 2 | 4 | True |
| OOM Error | | 4 | 2 | 4 | True |
| OOM Error | | 4 | 2 | 2 | True |
| OOM Error | | 2 | 2 | 2 | True |
| OOM Error | | 4 | 4 | 4 | True |
| OOM Error | | 4 | 4 | 2 | True |
| OOM Error | | 2 | 4 | 2 | True |
| OOM Error | | 4 | 4 | 8 | False |
| OOM Error | | 4 | 2 | 16 | False |
| OOM Error | | 4 | 2 | 8 | False |

| Step Time | MFU | MB Size | TP size | PP Size | Sequence Parallel |
|---|---|---|---|---|---|
| OOM Error | | 4 | 4 | 16 | False |
| OOM Error | | 4 | 4 | 4 | False |
| OOM Error | | 4 | 4 | 2 | False |
| OOM Error | | 2 | 2 | 4 | False |
| OOM Error | | 2 | 4 | 2 | False |
| OOM Error | | 1 | 2 | 2 | False |
| OOM Error | | 4 | 2 | 2 | False |
| OOM Error | | 2 | 2 | 2 | False |
| OOM Error | | 1 | 2 | 4 | False |
| OOM Error | | 2 | 4 | 4 | False |
| OOM Error | | 4 | 2 | 4 | False |
| OOM Error | | 1 | 4 | 2 | False |
| OOM Error | | 4 | 4 | 16 | False |
| OOM Error | | 4 | 2 | 16 | False |
| OOM Error | | 4 | 2 | 8 | False |
| OOM Error | | 4 | 4 | 8 | False |

## C.6  LLAMA 65B

Table 14: Performance analysis of a LLAMA 65B model trained on 64 A100 GPUs, with and without sequence parallelism. All measurements use FLASHATTENTION-2, the RMS norm kernel, and do not make use of activation checkpointing. The analysis also includes Out of Memory (OOM) error occurrences.

| Step Time | MFU | MB Size | TP size | PP Size | Sequence Parallel |
|---|---|---|---|---|---|
| 147.02 | 59.62 | 1 | 2 | 4 | True |
| 149.92 | 58.47 | 2 | 4 | 4 | True |
| 149.97 | 58.44 | 1 | 2 | 8 | True |
| 152.65 | 57.42 | 1 | 2 | 8 | False |
| 156.40 | 56.04 | 2 | 4 | 8 | True |
| 158.74 | 55.22 | 2 | 4 | 4 | False |
| 159.57 | 54.93 | 1 | 4 | 4 | True |
| 162.32 | 54.00 | 1 | 4 | 2 | True |
| 166.36 | 52.69 | 1 | 4 | 8 | True |
| 166.49 | 52.65 | 4 | 8 | 4 | False |
| 167.70 | 52.27 | 4 | 8 | 8 | False |
| 168.70 | 51.96 | 2 | 4 | 8 | False |
| 169.39 | 51.75 | 1 | 4 | 2 | False |
| 172.11 | 50.93 | 1 | 4 | 4 | False |
| 178.64 | 49.07 | 2 | 8 | 2 | True |
| 179.78 | 48.76 | 1 | 4 | 8 | False |
| 179.87 | 48.74 | 2 | 8 | 4 | True |
| 186.54 | 47.16 | 2 | 8 | 8 | True |
| 192.45 | 45.55 | 1 | 8 | 2 | True |
| 193.69 | 45.26 | 2 | 8 | 2 | False |
| 198.11 | 44.25 | 1 | 8 | 4 | True |
| 201.44 | 43.52 | 1 | 8 | 8 | True |
| 202.20 | 43.35 | 2 | 8 | 4 | False |
| 207.26 | 42.29 | 1 | 8 | 2 | False |
| 220.27 | 39.80 | 1 | 8 | 4 | False |
| 223.09 | 39.29 | 2 | 8 | 8 | False |
| 233.00 | 37.62 | 1 | 8 | 8 | False |
| OOM Error | | 4 | 4 | 2 | True |
| OOM Error | | 4 | 2 | 2 | True |
| OOM Error | | 4 | 2 | 4 | True |
| OOM Error | | 2 | 4 | 2 | True |
| OOM Error | | 2 | 2 | 8 | True |
| OOM Error | | 2 | 2 | 4 | True |
| OOM Error | | 2 | 2 | 2 | True |
| OOM Error | | 1 | 2 | 2 | True |
| OOM Error | | 2 | 2 | 2 | False |
| OOM Error | | 2 | 4 | 2 | False |
| OOM Error | | 2 | 2 | 4 | False |
| OOM Error | | 2 | 2 | 8 | False |
| OOM Error | | 1 | 2 | 4 | False |
| OOM Error | | 1 | 2 | 2 | False |
| OOM Error | | 4 | 8 | 2 | False |
| OOM Error | | 4 | 4 | 2 | False |
| OOM Error | | 4 | 4 | 8 | False |
| OOM Error | | 4 | 2 | 8 | False |
| OOM Error | | 4 | 4 | 4 | False |
| OOM Error | | 4 | 2 | 2 | False |
| OOM Error | | 4 | 2 | 4 | False |

