# OpenReview forum: "Efficient Parallelization Layouts for Large-Scale Distributed Model Training"
_NeurIPS.cc/2023/Workshop/WANT — WANT@NeurIPS 2023 Oral_

### Official Review · Reviewer_NFWW · 2023-10-19
**The manuscript provides a systematic analysis to find optimal distributed training configurations for efficiently training LLMs, including large models such as, LLAMA with 65bn parameters. More recent methods such as FLASHATTENTION have been incorporated in the analysis to shown an advancement to the state-of-the-art in terms of similar work. A wide range of configuration sweep is shown along with comparison of the achieved performance in terms of Model FLOPS Utilization (MFU) of the in-house framework with other established libraries.**

**Rating:** 7
**Confidence:** 3

**Review:**

Strengths:
The manuscript is well written, follows a clear structure and the findings are properly documented. The range of considered distributed trainings strategies is comprehensive and useful to guide training of LLMs.

Weakness:
The manuscript is missing some information, which will be clear if the questions (to follow) are answered and accordingly revised. In general, the comparison of the MFU with other established frameworks is not rigorous and the balance between tensor and pipeline parallelism is not well established.

Questions/comments:

•	Consistency in nomenclature is required. The authors introduced in l.69 that “…, we use model parallelism as an umbrella term for both tensor and pipeline parallelism”, but this doesn’t seem consistent with the analysis in Sec. 4.4 (“In contrast, our results favor pipeline parallelism over model parallelism”). This makes it difficult to evaluate the findings.

•	Furthermore, the reported findings from Fig. 4 are not conclusive, since the trend is changing for all models. This is also missing in the conclusions of the manuscript. One recommendation is to further increase the scope of configurations in Fig. 4 and perhaps add other models.

•	The MFU comparisons in Tab. 2 should consider same configurations, such as #GPUs, Batch Size, etc. If this is not possible, this should not be used as a comparison. The authors are advised to provide a consistent comparison.

•	One limitation of the paper is the maximum #GPUs that are considered in the manuscript, which (to repeat the point above) doesn’t allow a fair comparison. The performance is expected to degrade with higher parallelism. The authors should either show additional results with more #GPUs or clarify this point. What is the expected performance with higher number of GPUs, say in the order of thousands? The authors should consider adding scaling plots to show the effect of increase in the number of GPUs on performance.

•	And what is the expected impact of use of other hardware e.g. TPUs, or AMD-based systems? Has this been considered by the authors? If not, a comment on this and its expected impact on the obtained results should be mentioned.

•	The authors have not discussed about the communication library that has been used in the study. This information is important since the communication backend is instrumental in achieving higher parallel efficiency. Please include details on the communication library. Along these lines, a breakdown on the communication footprint based on the I/O, gradient calculation, etc. will be beneficial for readers.

Other issues:

•	l. 101: Perhaps rephrase to: “All trainings are conducted .. “

•	References are not properly cited – many are missing journal/conference names and volume

•	Font size in Fig. 1, 2 could be larger to improve readability. Fig. 3 also needs larger font size.

---

### Official Review · Reviewer_7fMA · 2023-10-23
**Interesting paper on LLM training, should be accepted**

**Confidence:** 5

**Review:**

This paper provides interesting insights on combining recent efficient training techniques with FlashAttention. There is no original algorithm proposed in the paper, but the reporting results can be helpful in practice.

Strengths:

1. Consider how useful FlashAttention is in LLM training, the recommendation settings of this paper can be very interesting to the people who tend to train a large model with the most recent efficient training techniques.
2. Combining these works seems non-trivial in practice, and the code to reproduce the experiments can certainly be useful (authors claimed to publish their code on GitHub).

Limitations:

2. Search space does not include ZERO-1,2,3, which can affect the effects of activation checkpointing/model parallelism.
3. The conclusions are mostly summarization of the phenomenons without deeper discussion. For instance, are the recommendation settings different from the prior works without recent techniques like FlashAttention?

---

### Official Review · Reviewer_hN1K · 2023-10-24
**This paper compares the performance of training efficiency for different Llama models and different combinations of parallelization strategies.**

**Confidence:** 5

**Review:**

This paper compares the performance of training efficiency for different Llama models and different combinations of parallelization strategies.


Overall, the paper is mainly experimental. Nevertheless, it compares the performance of different models and strategies in terms of training efficiency. In terms of parallel strategies, the paper is extremely rich, exploring data, tensor, pipeline, sequence paralellisms, themselves combined with checkpointing activation in cases where the parameters lead to excessive memory consumption on GPUs. The use of flash attention (an IO aware attention algorithm based on kernel fusion) is also studied, with the conclusion that FlashAttention-2 is always superior. In addition, the experiments are carried out on a platform of significant size, suitable for this type of models: 32 NVIDIA DGX nodes with 8 A100 80 GB memory and NVlink  with 600GB/s bandwidth on each node.

The paper is extremely informative, very relevant and at the heart of the WS themes, and will be of great interest to participants.

A limitation of the paper is that it does not contain any algorithmic contribution as such, and in particular does not offer an easy way to configure the various parameters (on 3D parallelism, micro-batch sizes, etc.). The paper relies on a brute-force exploration of a large number of configurations in order to choose the best ones, but it does provide some intuitions. Another limitation of the paper is that the validation is done on a single platform (though with different numbers of nodes), which makes it impossible to accurately assess the influence of different architectural parameters. Finally, comparisons between methods are sometimes weakened by the fact that they are not carried out on exactly the same configurations (it would be nice if this problem could be resolved for the final version of the paper).

Nevertheless, the above limitations do not question the paper's interest in WS WANT, and I recommend its publication.

Minor remarks
- Even a brief description of ZERO and FlashAttention would make the paper more self-contained and easier to read.
- Some pictures to illustrate how pipelined model parallelism works would also help. Perhaps the abstract should focus on quantitative results (the bulb explanation was not intuitive for me at this stage).
- I was unable to find any details on the framework (due to anonymization) used to perform the parallel execution and to configure the various parameters (not even in the appendix). In my opinion, it would be relevant to give this information clearly. Making the code available after the WS would also be very useful.

---

### Author Response · Authors · 2023-12-01

Dear Reviewers,

Thank you for your insightful feedback on our paper. We are pleased to address the points raised and have made revisions accordingly. We want to highlight the strengths of our work as mentioned in your reviews:
* **Comprehensive Analysis:** The range of distributed training strategies is extensive, providing valuable guidance for training large language models.
* **Insightful Findings:** The insights on combining efficient training techniques benefit practitioners in the field.

Key Revisions Made:
* **Nomenclature and Consistency:** We revised the paper to distinguish clearly between tensor and pipeline parallelism.
* **We added a Limitation Section:** We discuss various limitations addressed by the reviewer, such as the applicability of recommendations to other frameworks, global batch size, number of GPUs, and the transferability to different model architectures and accelerators.
* **Minor Corrections:** We have addressed minor issues, including phrasing, citation, figure readability, inconsistencies in Section 4.4, and information about the communication library.

We would also like to point out that we only promised to publish the data and the scripts to reproduce the figures based on the data in our original submission.

We appreciate the opportunity to improve our paper with your feedback.
Thank you for your consideration and valuable input.

---

### Meta-Review · Area_Chair_ED2x · 2023-10-27

**Recommendation:** Accept (Oral)
**Confidence:** 4

**Metareview:**

Very informative experimental paper relevant to the workshop. While the paper is mainly experimental and doesn't introduce novel algorithms, all reviewers agree that it is highly informative from a practical point of view and recommended accepting the paper. I therefore recommend it for an Oral.

---

### Decision · Program_Chairs · 2023-10-28

**Decision:**

Accept (Oral)

**Comment:**

We thank the authors for their time and contribution to WANT and we are pleased to share that after the reviewing process the paper has been accepted. Congratulations! We encourage the authors to consider reviewers' feedback for the improvement of the camera-ready version. We hope to see you in person at the workshop and brainstorm on efficient training research together!